# Dissociating orexin-dependent and -independent functions of orexin neurons using novel Orexin-Flp knock-in mice

Srikanta Chowdhury[1,2,3†], Chi Jung Hung[1,2,3], Shuntaro Izawa[1,2,3], Ayumu Inutsuka[1], Meiko Kawamura[4], Takashi Kawashima[5], Haruhiko Bito[5], Itaru Imayoshi[6], Manabu Abe[4], Kenji Sakimura[4], Akihiro Yamanaka[1,2,3]*

[1]Department of Neuroscience II, Research Institute of Environmental Medicine, Nagoya University, Nagoya, Japan; [2]Department of Neural Regulation, Graduate School of Medicine, Nagoya University, Nagoya, Japan; [3]CREST, JST, Honcho Kawaguchi, Saitama, Japan; [4]Department of Animal Model development, Brain Research Institute, Niigata University, Niigata, Japan; [5]Department of Neurochemistry, Graduate School of Medicine, The University of Tokyo, Tokyo, Japan; [6]Research Center for Dynamic Living Systems, Graduate School of Biostudies, Kyoto University, Kyoto, Japan

*For correspondence:
yamank@riem.nagoya-u.ac.jp

Present address: †Department of Biochemistry and Molecular Biology, University of Chittagong, Chittagong, Bangladesh

Competing interests: The authors declare that no competing interests exist.

**Abstract** Uninterrupted arousal is important for survival during threatening situations. Activation of orexin/hypocretin neurons is implicated in sustained arousal. However, orexin neurons produce and release orexin as well as several co-transmitters including dynorphin and glutamate. To disambiguate orexin-dependent and -independent physiological functions of orexin neurons, we generated a novel Orexin-flippase (Flp) knock-in mouse line. Crossing with Flp-reporter or Cre-expressing mice showed gene expression exclusively in orexin neurons. Histological studies confirmed that orexin was knock-out in homozygous mice. Orexin neurons without orexin showed altered electrophysiological properties, as well as received decreased glutamatergic inputs. Selective chemogenetic activation revealed that both orexin and co-transmitters functioned to increase wakefulness, however, orexin was indispensable to promote sustained arousal. Surprisingly, such activation increased the total time spent in cataplexy. Taken together, orexin is essential to maintain basic membrane properties and input-output computation of orexin neurons, as well as to exert awake-sustaining aptitude of orexin neurons.
DOI: https://doi.org/10.7554/eLife.44927.001

## Introduction

Orexin A (hypocretin-1) and orexin B (hypocretin-2) (*de Lecea et al., 1998*; *Sakurai et al., 1998*), generated from the same precursor protein called prepro-hypocretin (encoded by *hypocretin* gene), are endogenous ligands for two closely related G-protein-coupled receptors termed orexin-1 receptor (OX1R) and orexin-2 receptor (OX2R) (*Sakurai et al., 1998*). Although orexin was named for its effect on inducing feeding behavior, it gained immense interest in sleep research as the knockout (KO) of prepro-orexin or dysfunction of OX2R in canines or mice reportedly mimics the human sleep disorder narcolepsy (*Chemelli et al., 1999*; *Lin et al., 1999*). Narcolepsy is a neurological disorder characterized by fragmented sleep/wakefulness, persistent daytime sleepiness and brief episodes of muscle weakness called cataplexy, which is often triggered by positive emotions. Narcolepsy patients showed a loss of overall hypocretin mRNA, confirming the association between orexin neuronal loss and the pathogenesis of narcolepsy (*Peyron et al., 2000*; *Thannickal et al., 2000*).

A small number of orexin-producing neurons (orexin neurons) are exclusively distributed in the lateral hypothalamic area (LHA) and perifornical area, but send projections widely throughout the major brain areas (*Nambu et al., 1999*; *Peyron et al., 1998*). Thus, it is no surprise that orexin neurons have diverse physiological roles, including the regulation of sleep/wakefulness (*Chemelli et al., 1999*; *Lin et al., 1999*), energy homeostasis (*Yamanaka et al., 2003*), thermoregulation (*Tupone et al., 2011*), as well as regulation of heart rate and blood pressure (*Zhang et al., 2006*). Using optogenetic and/or chemogenetic techniques, we and others showed that orexin neurons modulate sleep/wake cycles in rodents (*Adamantidis et al., 2007*; *Sasaki et al., 2011*; *Tsunematsu et al., 2011*). In addition to these, employing transgenic mice, we reported that chemogenetic activation of orexin neurons increased locomotion, feeding behavior and metabolism (*Inutsuka et al., 2014*). Thus, orexin neurons are thought to interact with the neuroregulatory, autonomic and neuroendocrine systems, and perform critical roles in the regulation of sleep/wakefulness and energy homeostasis.

The physiological activity of orexin neurons is modulated by multiple neural inputs and humoral factors. These inputs include GABAergic neurons in the preoptic area, serotonergic neurons in the dorsal and median raphe nuclei, central amygdala, basal forebrain cholinergic neurons, the bed nucleus of the stria terminalis, supraventricular zone, and the dorsomedial, lateral and posterior hypothalamus (*Sakurai et al., 2005*; *Yoshida et al., 2006*). Recently, we reported that serotonergic neurons in the raphe nucleus inhibit orexin neurons both directly and indirectly (*Chowdhury and Yamanaka, 2016*). Orexin neurons are also found to respond to multiple humoral factors and neuropeptides (*Inutsuka and Yamanaka, 2013*; *Sakurai, 2014*). Most interestingly, orexin neurons form a positive-feedback circuitry among themselves in the LHA by activating other orexin neurons through OX2R to maintain the wake-active network at optimum level and/or for a sustained period (*Yamanaka et al., 2010*).

However, orexin neurons contain other neurotransmitters including glutamate (*Rosin et al., 2003*), dynorphin (*Chou et al., 2001*) and galanin (*Håkansson et al., 1999*). Therefore, to better understand the physiology and pathophysiology of orexin/hypocretin system, it is essential to disambiguate the roles of orexin from other co-transmitters. To address this, we generated novel Orexin-flippase (Flp) knock-in (KI) mice, which express Flp recombinase under control of *hypocretin* gene in mice. Employing this novel mouse line, we found that orexin performs critical roles in maintaining basic electrophysiological properties of orexin neurons as well as to initiate feed-forward activation of orexin neurons by facilitating excitatory glutamatergic inputs. Focusing on sleep/wakefulness, we also manipulated orexin neuronal activity and evaluated their physiological effects in freely-moving mice. To achieve this, we employed a chemogenetic technique, Designer Receptors Exclusively Activated by Designer Drugs (DREADD) (*Armbruster et al., 2007*). Our results showed orexin plays a major role in the awake-sustaining aptitude of orexin neurons. Although the phenotypic consequences of orexin neurons activation in increased wakefulness were reinforced for by co-transmitters, these co-transmitters rather deteriorated cataplexy. Together, these data clearly identified the importance of orexin neuropeptides for the physiological function of orexin neurons and their postsynaptic partners.

## Results

### Generation of novel Orexin-Flp (OF) mice

We previously generated Orexin-Cre (*Inutsuka et al., 2014*) or orexin-tetracycline-controlled transactivator (Orexin-tTA) (*Tabuchi et al., 2013*) transgenic mice in which Cre recombinase or tTA was expressed in orexin neurons to allow regulation of transgene expression in the orexin neurons. However, expression of Cre or tTA in those transgenic mice was driven by a 3.2 kb short gene fragment upstream of the human *hypocretin* gene. Thus, we generated a new mouse by targeting the *hypocretin* gene of the mouse genome. Employing a homologous recombination system, here we generated Orexin-Flp (OF) mice by knocking in the EGFP-2A-Flp transgene just downstream of the translation initiation site of the *hypocretin* gene in-frame (*Figure 1A*). To visualize the function of Flp recombinase, we bred these mice with Flp-reporter (FSF-mTFP1) mice (*Imayoshi et al., 2012*) in which fluorescent protein mTFP1 is expressed in a Flp-dependent manner (*Figure 1B*). Immunohistochemical analysis showed mTFP expression in 80.9 ± 4.4% of orexin-immunoreactive neurons. We

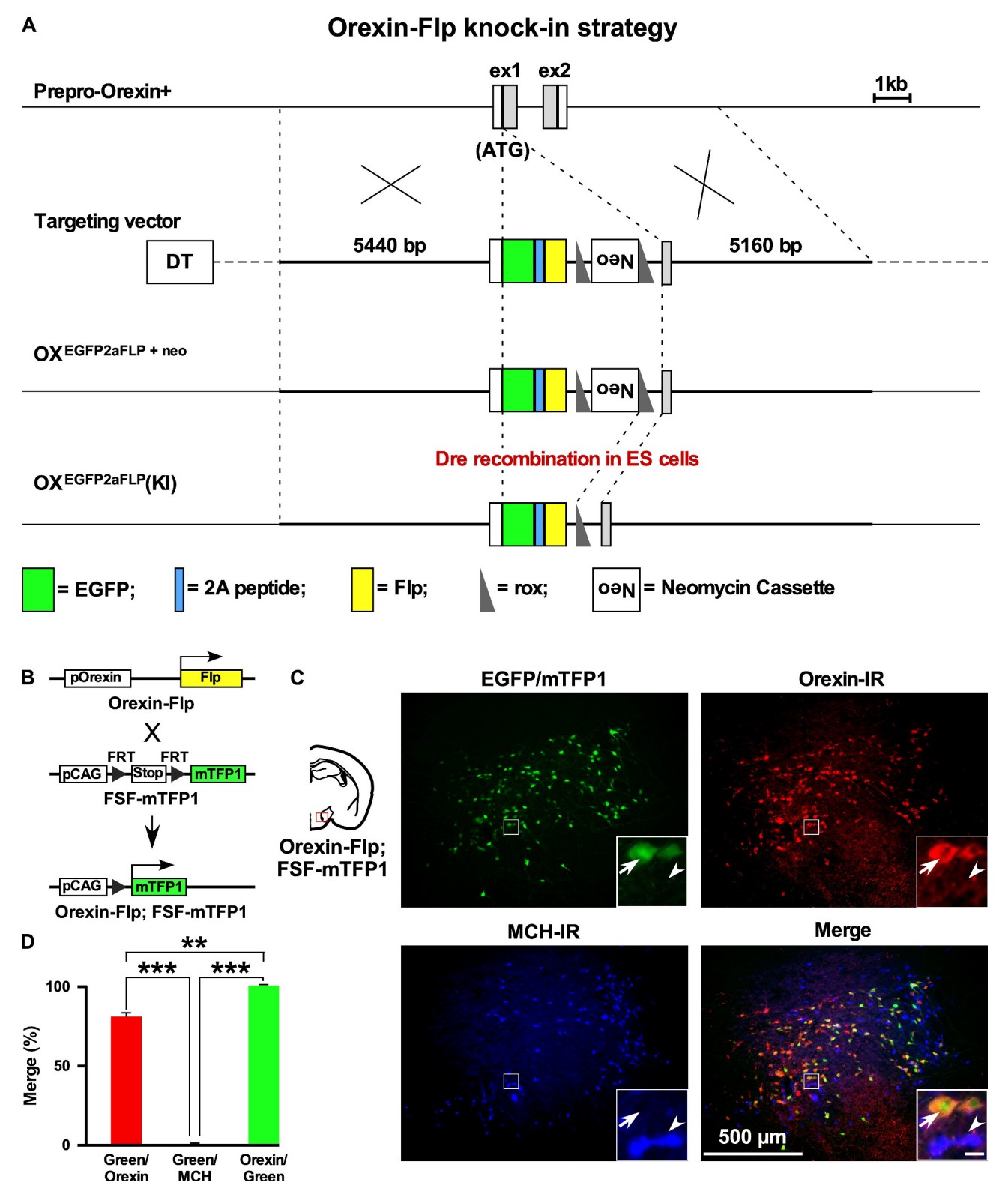

**Figure 1.** Generation of OF mice. (**A**) Schematic representations of the *hypocretin* gene, targeting vector, and targeted gene. To achieve orexin neuron-specific expression of Flp recombinase, we inserted EGFP-2A-Flp just behind the translation initiation site of the *hypocretin* gene in-frame. Viral T2A peptide is cleaved just after translation, and EGFP and Flp recombinase localize independently. DT, diphtheria toxin; Neo, neomycin-resistant gene expression cassette. (**B**) Structure of the reporter gene in the presence of Flp. Orexin-Flp; FSF-mTFP1 bigenic mice were generated to express

*Figure 1 continued on next page*

*Figure 1 continued*

mTFP1 in orexin neurons. (**C**) Representative pictures from coronal brain sections of Orexin-Flp; FSF-mTFP1 bigenic mice. Arrow indicates mTFP1 and/ or EGFP expressing orexin neurons and the arrowhead indicates the position of the MCH neuron. Orexin-IR, orexin-immunoreactive; MCH-IR, MCH-immunoreactive. Inset scale bar: 20 µm. (**D**) Summary of the co-expression analysis (n = 4 mice). The p values were determined using one-way ANOVA test followed by a post-hoc Tukey analysis. Data represent the mean ± SEM.

DOI: https://doi.org/10.7554/eLife.44927.002
The following source data is available for figure 1:

**Source data 1.** Source data for *Figure 1D*.
DOI: https://doi.org/10.7554/eLife.44927.003

observed no ectopic expression of EGFP and/or mTFP1 protein in melanin-concentrating hormone (MCH)-immunoreactive (n = 4 mice; *Figure 1C and D*) or in any other non-orexin-immunoreactive neurons in these bigenic mice. In other words, all reporter gene-expressing neurons were orexin-immunoreactive (IR). Together, these data confirmed that functional Flp was expressed exclusively in the orexin neurons of newly generated OF mice.

## OF mice enable the dual targeting of adjacent cell types

By taking advantage of the vast repertoire of cell type-specific Cre- or tTA-expressing mice, the new OF mice may enable the expression of various genes in different subsets of neurons (*Figure 2A*). To confirm this, we next generated two different bigenic mice by crossing OF mice with either MCH-Cre (encoded by *Pmch* gene) or Gad67-Cre (encoded by *Gad1* gene) mice (*Figure 2B and E*). MCH-Cre or Gad67-Cre mice exclusively express Cre recombinase in MCH (*Higo et al., 2009*) or GABA neurons (*Kong et al., 2010*), respectively. To test whether we could target dual cell types simultaneously inside the LHA, we performed injection of an AAV cocktail composed of an equal volume of AAV(DJ)-CMV-FLEX-tdTomato and AAV(DJ)-CMV-dFRT-hrGFP in both Orexin-Flp; MCH-Cre and Orexin-Flp; Gad67-Cre bigenic mice (*Figure 2C and F*). Given that MCH and GABA neurons are distinct neuronal subsets from orexin neurons in the LHA (*Broberger et al., 1998*; *Rosin et al., 2003*), we expected a discrete expression of tdTomato and hrGFP in both bigenic mice. Indeed, we observed that Flp and Cre recombinase-driven fluorescent protein expression did occur in distinct populations (*Figure 2D and G*) in coronal brain sections (n = 3 mice for each). These data demonstrated that a broader range of opportunities for cell type-specific manipulation and/or activity recording is possible with OF mice.

## OF (KI/KI) homozygote knockout orexin peptides

Since the EGFP-2A-Flp sequence was inserted in-frame at the start codon of the *hypocretin* gene using the KI method, we reasoned that KI/KI homozygous mice are essentially orexin KO mice. To confirm, immunohistochemical studies were conducted to compare the expression of EGFP, orexin and dynorphin in the LHA of heterozygous OF (KI/-) and homozygous OF (KI/KI) mice (*Figure 3A and B*). As expected, we found that EGFP expressing neurons were distributed in the LHA, however, orexin-immunoreactivity was absent in the LHA in OF (KI/KI) homozygous mice (*Figure 3B*). In heterozygous OF (KI/-) mice, 41.1 ± 5.2% of orexin-positive neurons were colocalized with EGFP (*Figure 3—figure supplement 1B*). This value was lower than results obtained from reporter mice (80.9 ± 4.4%), and is likely due to detection limitations of EGFP fluorescence due to low levels of expression. We stained every 4[th] slice of mouse brain containing LHA and counted 567 ± 25 orexin-positive and 673 ± 55 dynorphin-positive cells/animal in heterozygous mice (n = 3 mice). Among the counted neurons, 84.9 ± 4.4% of dynorphin-positive neurons co-expressed orexin (*Figure 3—figure supplement 1A*). However, while we counted 661 ± 15 dynorphin-positive cells/animal, we did not find any orexin-positive neurons in the homozygous mice (n = 3 mice, Hetero vs Homo, p=0.847 (dynorphin), p=2.4e-5 (orexin), unpaired *t*-test), showing that the number of dynorphin-positive cells were comparable to that in heterozygous animals. These immunohistochemical studies confirmed that orexin neurons in homozygous mice did not express orexin, however, expression of dynorphin was unaffected.

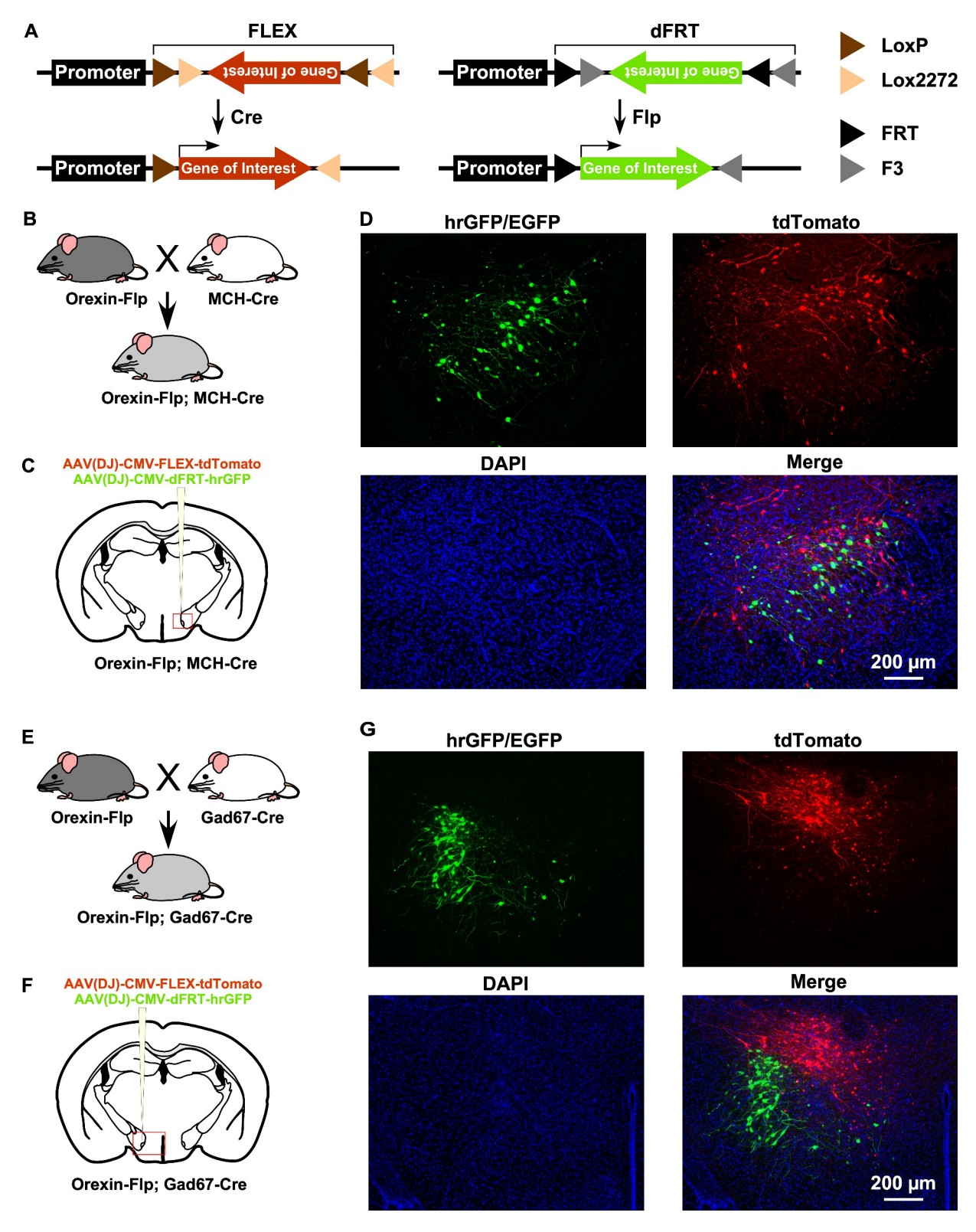

**Figure 2.** Gene expression control using OF mice to target different cell types within the same brain region. (**A**) Schematic showing Cre (left) and Flp (right) recombinase-dependent gene expression control. (**B and E**) The breeding scheme for Orexin-Flp; MCH-Cre and Orexin-Flp; Gad67-Cre bigenic mice, respectively. (**C and F**) Schematic drawings showing micro-injection of the AAV cocktail into the LHA of bigenic mice. (**D and G**) Representative coronal brain sections showing the segregated expression of tdTomato and hrGFP in a Cre and Flp recombinase-dependent manner, respectively.

*Figure 2 continued on next page*

*Figure 2 continued*

DOI: https://doi.org/10.7554/eLife.44927.004

## Electrophysiological properties of orexin neurons lacking orexin

The expression of EGFP in the OF mice allowed us to visualize orexin neurons in the acute brain slice preparations. To this end, we sought to evaluate the importance of orexin neuropeptides in conserving the electrophysiological properties of orexin neurons. Therefore, we recorded and compared resting membrane potentials (Vrest), firing frequency, input resistance and capacitance in OF (KI/-), OF (KI/KI), and Orexin-EGFP (Tg/Tg) (OE) (*Yamanaka et al., 2003*) mice (*Figure 4A–4E*). Orexin neurons in the OF (KI/KI) mice were found to have significantly hyperpolarized membrane potential (−58.8 ± 1.2 mV; n = 21 cells) compared to OF (KI/-) mice (−51.6 ± 0.9 mV; n = 21 cells, p=1.0e-5) and OE mice (−50.4 ± 0.9 mV; n = 22 cells, p=3.5e-7; *Figure 4A–i–4A-iv*). Spontaneous firing frequency measured by cell-attached recordings were also found to be significantly lower in orexin neurons in OF (KI/KI) mice (1.5 ± 0.2 Hz; n = 25 cells) compared to OF (KI/-) mice (2.5 ± 0.3 Hz; n = 21 cells, p=0.015) and OE mice (2.6 ± 0.2 Hz; n = 25, p=0.004; *Figure 4B–i–4B-iv*). This lower discharge rate could be attributed to the hyperpolarized membrane potential of orexin neurons in homozygous mice. We next measured the input resistance of identified neurons by measuring the voltage deviation generated by current injection from −100 to +100 pA in the current clamp protocol. In mice lacking orexin peptide (OF (KI/KI) mice), orexin neurons were found to have significantly lower input resistance (485.9 ± 35.8 MΩ; n = 23 cells) compared to OF (KI/-) mice (639.9 ± 48.2 MΩ; n = 21 cells, p=0.037) and OE mice (638.2 ± 43.4 MΩ; n = 24 cells, p=0.032; *Figure 4C–i–4C-iv*). The lower input resistance in neurons lacking orexin peptides suggests that greater synaptic inputs (current injection) are necessary to generate changes in membrane potential. We also compared the membrane capacitance measured during whole-cell recordings. Interestingly, we observed that orexin neurons in OF (KI/KI) mice had significantly higher membrane capacitance (35.4 ± 1.3 pF; n = 25 cells) than OF (KI/-) mice (29.9 ± 1.5 pF; n = 25 cells, p=0.012) and OE mice (30.0 ± 1.3 pF; n = 25 cells, p=0.014, one-way ANOVA post-hoc Tukey; *Figure 4E*). This higher capacitance in orexin neurons that lack orexin peptides reflects the increased surface area of the plasma membrane. Taken together, these data clearly suggest that orexin neuropeptide is essential for maintaining the active and passive electrophysiological properties of orexin neurons.

## Orexin mediates feed-forward activation of orexin neurons via facilitation of glutamatergic inputs

To characterize whether the excitatory and inhibitory synaptic inputs onto orexin neurons are regulated by the presence or absence of orexin peptides, we recorded glutamatergic and GABAergic inputs using the voltage clamp method. We recorded spontaneous excitatory post-synaptic currents (sEPSCs) from EGFP-expressing orexin neurons in the presence of picrotoxin (400 μM), a GABA-A receptor antagonist. We found that orexin neurons lacking orexin protein had significantly lower sEPSC frequency while the amplitudes were unaffected (*Figure 5A–5E*). The average inter-event interval for all recorded sEPSCs of orexin neurons in OF (KI/KI) mice was 247.2 ± 48.6 ms (n = 29 cells) while that of OF (KI/-) mice was 116.7 ± 7.6 ms (n = 25 cells, p=0.01 vs OF (KI/KI)) and OE mice was 77.9 ± 7.4 ms (n = 26 cells, p=5.8e-4 vs OF (KI/KI), one-way ANOVA pot-hoc Tukey; *Figure 5D*). The sEPSC amplitude of orexin neurons in OF (KI/KI) mice was 22.6 ± 1.3 pA (n = 29 cells) while that of OF (KI/-) mice was 22.6 ± 1.3 pA (n = 25 cells, p=1.0 OF (KI/KI)) and in OE mice was 20.5 ± 1.1 pA (n = 26 cells, p=0.46 vs OF (KI/KI), one-way ANOVA post-hoc Tukey; *Figure 5E*).

Next, we recorded spontaneous inhibitory post-synaptic currents (sIPSCs) from orexin neurons in the presence of CNQX (20 μM) and AP-5 (50 μM) to block glutamatergic inputs. Although sIPSCs in neurons lacking orexin peptides showed a tendency toward decreasing frequency and amplitude, these changes were not statistically significant. The inter-event interval for sIPSCs in orexin neurons in OF (KI/KI) mice was 835.0 ± 124.6 ms (n = 17 cells) while that of OF (KI/-) mice was 642.4 ± 70.9 ms (n = 22 cells, p=0.35 vs OF (KI/KI)) and in OE mice was 595.0 ± 96.7 ms (n = 21 cells, p=0.21 vs OF (KI/KI), one-way ANOVA post-hoc Tukey; *Figure 5H*). The sIPSC amplitude in orexin neurons in OF (KI/KI) mice was 48.4 ± 6.1 pA (n = 17 cells) while that of OF (KI/-) mice was 52.5 ± 3.9 pA (n = 22 cells, p=0.86 vs OF (KI/KI)) and in OE mice was 57.7 ± 6.5 pA (n = 21 cells, p=0.48 vs OF (KI/

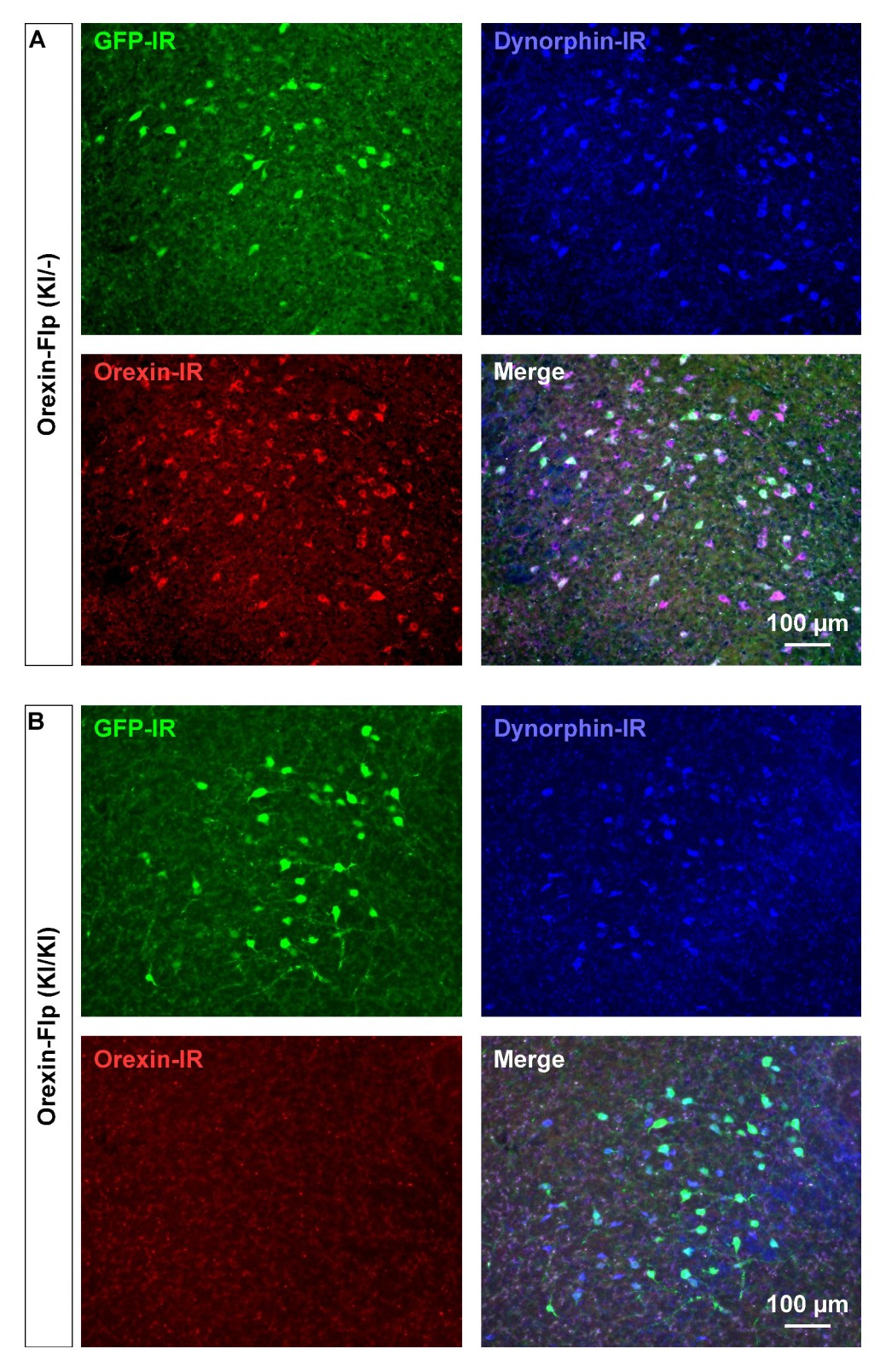

**Figure 3.** Immunohistochemical confirmation of OF mice. (**A and B**) Representative coronal brain sections showing the expression of EGFP (green), orexin (red) and dynorphin (blue). GFP-IR, GFP-immunoreactive; Dynorphin-IR, Dynorphin-immunoreactive; Orexin-IR, orexin-immunoreactive.
DOI: https://doi.org/10.7554/eLife.44927.005

*Figure 3 continued on next page*

*Figure 3 continued*

The following source data and figure supplements are available for figure 3:

**Figure supplement 1.** Immunohistochemical expression analysis in OF (KI/-) and OF (KI/KI) mice.
DOI: https://doi.org/10.7554/eLife.44927.006
**Figure supplement 1—source data 1.** Source data for *Figure 3—figure supplement 1A and B*.
DOI: https://doi.org/10.7554/eLife.44927.007

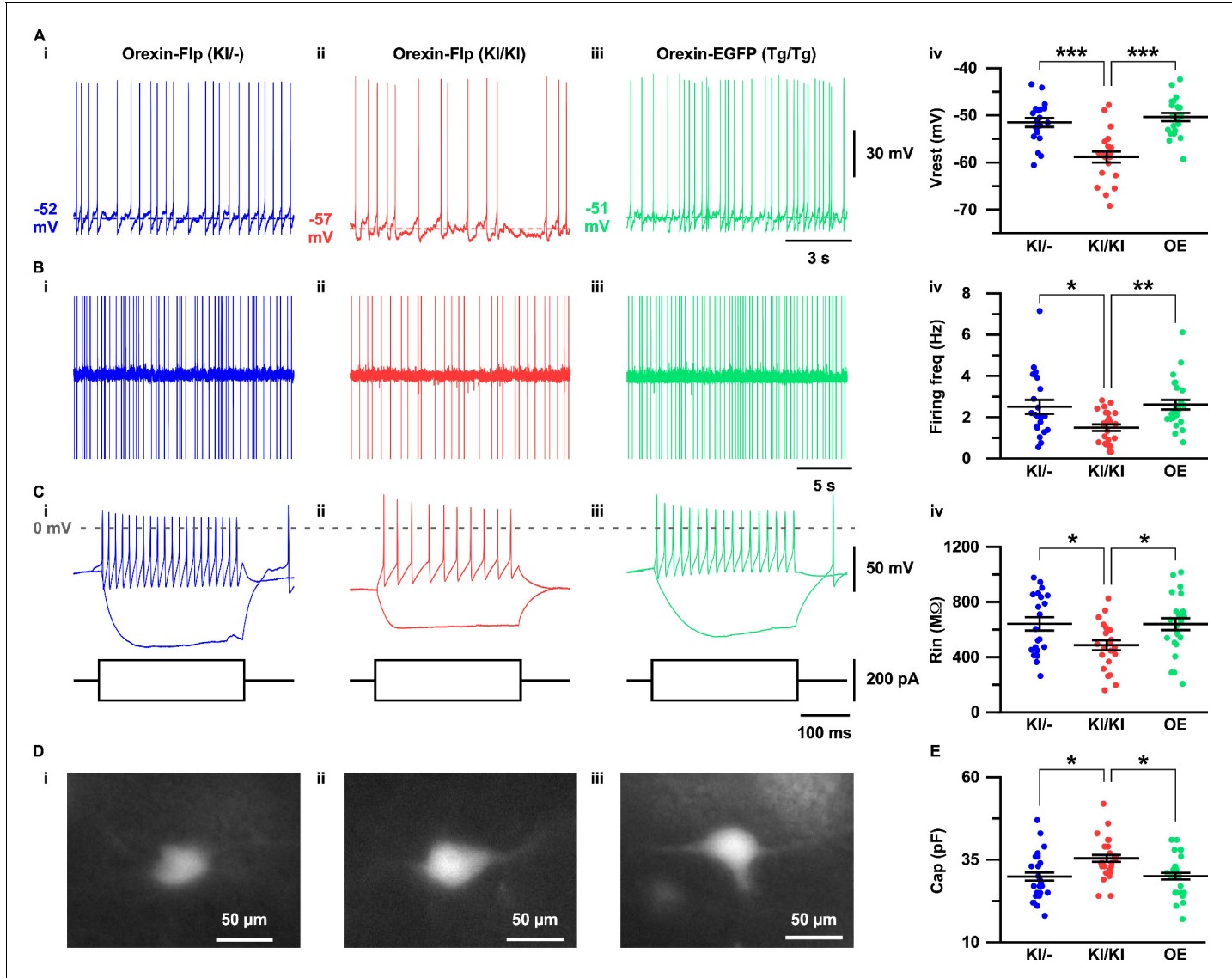

**Figure 4.** Electrophysiological properties of orexin neurons with/without orexin. (A-C) show representative traces recorded from OF (KI/-) (i), OF (KI/KI) (ii) and OE (iii) mice. (A) Membrane potential in whole-cell current clamp recordings. (B) Spontaneous firing in cell-attached recordings. (C) Step current injection-induced membrane potential changes. Panel iv is a summary of the data in panel i to iii. (D) Representative images showing EGFP expression in acute coronal brain slices during electrophysiological recording. (E) Cell capacitance from whole-cell current clamp recording. The p values were determined using one-way ANOVA test followed by a post-hoc Tukey analysis. Data represent the mean ± SEM.
DOI: https://doi.org/10.7554/eLife.44927.008

The following source data is available for figure 4:

**Source data 1.** Source data for *Figure 4Aiv, 4Biv, 4Civ and 4E*.
DOI: https://doi.org/10.7554/eLife.44927.009

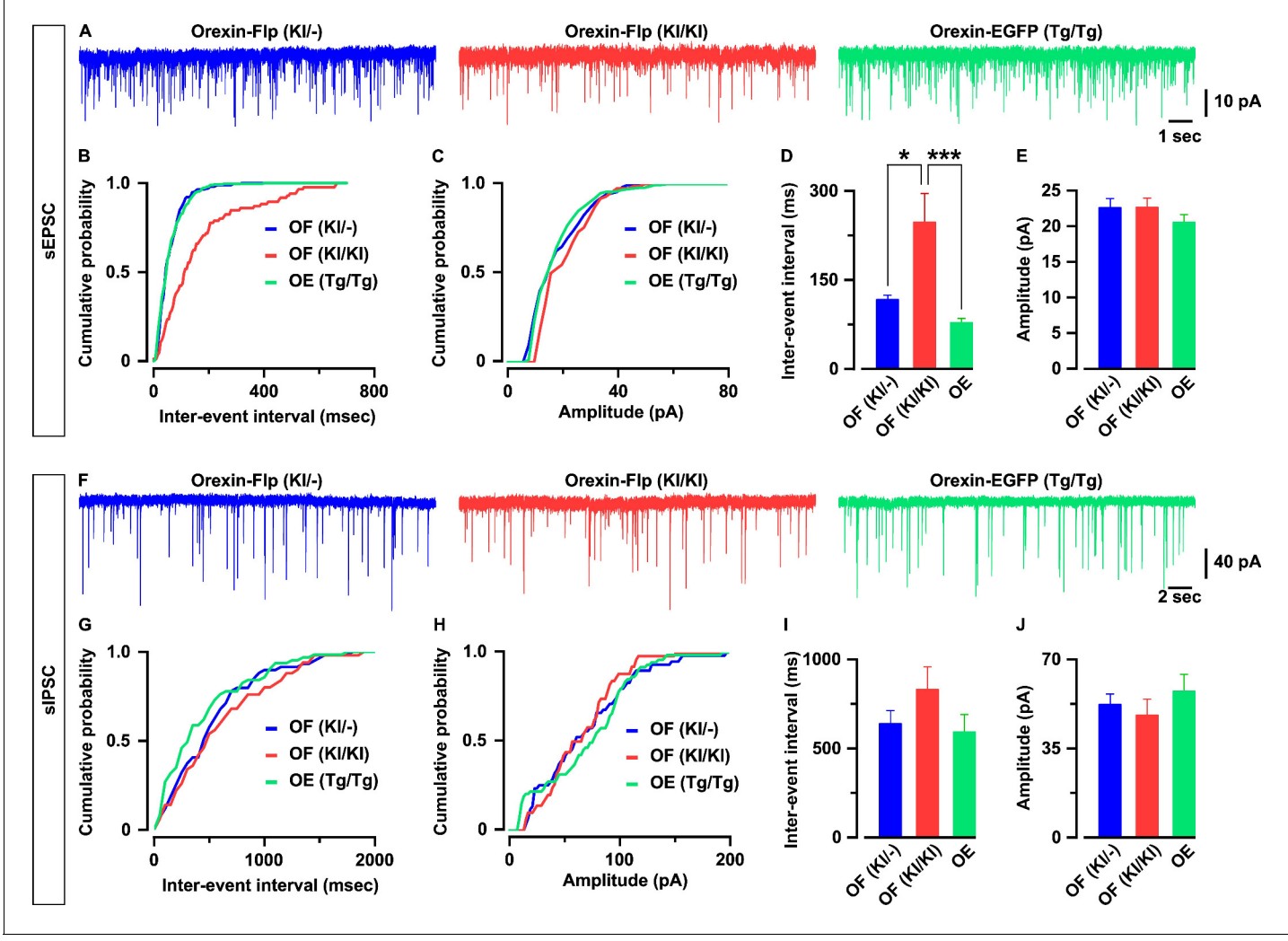

**Figure 5.** Orexin neurons receive fewer glutamatergic inputs in the absence of orexin. (A) Representative sEPSC traces recorded from EGFP-expressing neurons at a holding potential of −60 mV. (B-C) Cumulative probability plot for the representative traces shown in A. Bar diagrams in D and E summarize the sEPSC data. (D) Inter-event interval. (E) amplitude (n = 25–29 cells). F, Representative sIPSC traces recorded from EGFP-expressing neurons at a holding potential of −60 mV. (G-H) Cumulative probability plot for the representative traces shown in F. Bar diagrams in (I and J) summarize the sIPSC data. (G) Inter-event interval. (H) amplitude (n = 17–22 cells). The p values were calculated by one-way ANOVA followed by a post-hoc Tukey test. Data represent the mean ± SEM.

DOI: https://doi.org/10.7554/eLife.44927.010
The following source data is available for figure 5:

**Source data 1.** Source data for *Figure 5D, E and I, J*.
DOI: https://doi.org/10.7554/eLife.44927.011
**Source data 2.** Source data for *Figure 5D, E and I, J*.
DOI: https://doi.org/10.7554/eLife.44927.012

KI), one-way ANOVA post-hoc Tukey; *Figure 5I*). These, along with previous, results indicate that orexin plays a role in maintaining the physiological input-output functions in orexin neurons.

## OF (KI/KI) mice showed symptoms in narcolepsy

We reasoned that if orexin is successfully knocked out from OF (KI/KI) mice, it must show the sign of behavioral arrest, cataplexy, which is defined as the sudden and reversible episodes of the drop of voluntary muscle tone while remains fully conscious during the episodes (*Tabuchi et al., 2014*). Thus, we recorded and analyzed the baseline sleep/wakefulness cycle in OF (KI/KI) mice. Behavioral

states shown by OF (KI/KI) mice were classified in four states which includes either wakefulness, REM, NREM or cataplexy (see Materials and methods). All recorded OF (KI/KI) mice showed cataplexy attack, especially during the start of the dark period. We also compared the vigilance state parameters of the OF (KI/KI) mice with those of wild type (WT) control mice and the values are presented in *Table 1*. As expected, OF (KI/KI) mice showed less wakefulness and higher NREM and REM sleep compared to the WT control animals (*Table 1*). Along with these changes, orexin KO mice showed an increased number of bouts and decreased average duration in all vigilance states (*Table 1*), suggesting fragmentation of the sleep-wakefulness and an inability to maintain sustained arousal in the absence of orexin peptides. However, in comparison to the previously generated orexin KO (*Chemelli et al., 1999*) as well as to the orexin neuron-ablated mice (*Hara et al., 2001*), newly generated OF (KI/KI) mice showed higher REM sleep and wakefulness and a lower amount of NREM sleep. This is presumably due to changes in the recording environment, different criteria of the definition of vigilance state and/or changes in the genetic background as the WT control mice show similar changes.

## Chemogenetic activation of orexin neurons lacking orexin neuropeptides

Next, we sought to isolate the physiological effects of orexin from those of all other neurotransmitters co-released by orexin neurons. To achieve this, we employed chemogenetics, DREADD. Using Flp recombinase-dependent gene expression, hM3Dq was exclusively expressed in orexin neurons. Here, hM3Dq was fused with mCherry to detect expression and localization. AAV(9)-CMV-dFRT-hM3Dq-mCherry was injected bilaterally into the LHA of both homozygous OF (KI/KI) and heterozygous OF (KI/-) mice (*Figure 6A*). We quantified the number of orexin-positive cells in every 4[th] brain slices. The number of orexin-immunoreactive cells was 439 ± 23 cells/mouse in the brain of heterozygous mice and, among these, 92.1 ± 0.6% expressed mCherry (n = 3 mice; *Figure 6B*). Moreover, a very similar number of LHA neurons expressed mCherry in both heterozygous and homozygous mice: 422 ± 80 cells/animal (n = 6 mice) in OF (KI/-) and 455 ± 74 cells/animal (n = 6 mice) in OF (KI/KI) mice expressed mCherry (p=0.76, unpaired *t*-test). To confirm the selective activation of hM3Dq-expressing neurons in vivo, we measured the expression of an immediate early gene product, c-Fos, which is a surrogate molecular marker of neuronal activity. Following the behavioral studies, six randomly selected mice from both the heterozygous and homozygous groups received i.p. administration of either saline or clozapine N-Oxide (CNO) (1.0 mg/kg). Animals were perfused, and tissues were collected 90 min after the injection. c-Fos staining of brain slices confirmed that CNO selectively activated hM3Dq-expressing neurons in both heterozygous and homozygous mice. In heterozygous mice, the ratio of c-Fos expression in mCherry-positive cells of saline or CNO injected mice

**Table 1.** Vigilance state parameters recorded from OF (KI/KI) and WT animals.

The table shows total time spent in individual states in minutes, duration of respective states in seconds and number of episodes (bouts) observed in either 24 hr, or only in light or dark period in OF (KI/KI, n = 7 mice; WT, n = 8 mice). *, p<0.05. The p values were determined using two-tailed unpaired student's *t*-test. Values are represented as the mean ± SEM.

| | | REM | | Cataplexy | | NREM | | Wake | |
|---|---|---|---|---|---|---|---|---|---|
| | | Orexin-Flp (KI/KI) | WT | Orexin-Flp (KI/KI) | WT | Orexin-Flp (KI/KI) | WT | Orexin-Flp (KI/KI) | WT |
| 24 hr | Total time (min) | 96.0 ± 5.7 | 92.2 ± 4.6 | 18.7 ± 6.3 | | 584.6 ± 19.1 | 555.7 ± 14.8 | 740.7 ± 18.3 | 792.1 ± 19.1 |
| | Duration (sec) | 55.7 ± 4.2 | 57.3 ± 2.4 | 61.5 ± 12.1 | | 81.4 ± 7.0 | 94.8 ± 5.8 | 102.1 ± 8.4* | 189.9 ± 16.8 |
| | Bouts | 110.7 ± 8.5 | 96.8 ± 7.5 | 17.4 ± 5.2 | | 442.3 ± 37.4* | 359.3 ± 20.8 | 453.0 ± 36.4* | 358.6 ± 20.8 |
| Light period | Total time (min) | 57.3 ± 3.8* | 68.4 ± 2.9 | 2.1 ± 0.8 | | 355.1 ± 13.5* | 390.8 ± 4.9 | 305.5 ± 14.6* | 260.8 ± 7.2 |
| | Duration (sec) | 48.4 ± 3.9* | 58.9 ± 1.8 | 48.9 ± 13.0 | | 93.4 ± 7.5 | 94.6 ± 5.0 | 80.1 ± 4.6* | 63.1 ± 4.2 |
| | Bouts | 73.4 ± 7.6 | 70.0 ± 4.4 | 2.0 ± 0.7 | | 234.6 ± 17.3 | 251.8 ± 14.1 | 231.0 ± 17.6 | 251.1 ± 13.9 |
| Dark period | Total time (min) | 38.7 ± 4.8* | 23.8 ± 2.5 | 16.6 ± 5.6 | | 229.5 ± 12.4* | 164.9 ± 10.5 | 435.2 ± 11.3* | 531.3 ± 12.7 |
| | Duration (sec) | 63.0 ± 6.2 | 55.6 ± 4.0 | 74.1 ± 19.8 | | 69.3 ± 8.2* | 94.9 ± 7.6 | 124.0 ± 12.7* | 316.6 ± 30.9 |
| | Bouts | 37.3 ± 4.3* | 26.8 ± 3.5 | 15.4 ± 4.7 | | 207.7 ± 21.1* | 107.5 ± 10.5 | 222.0 ± 20.6* | 107.5 ± 10.6 |

DOI: https://doi.org/10.7554/eLife.44927.013

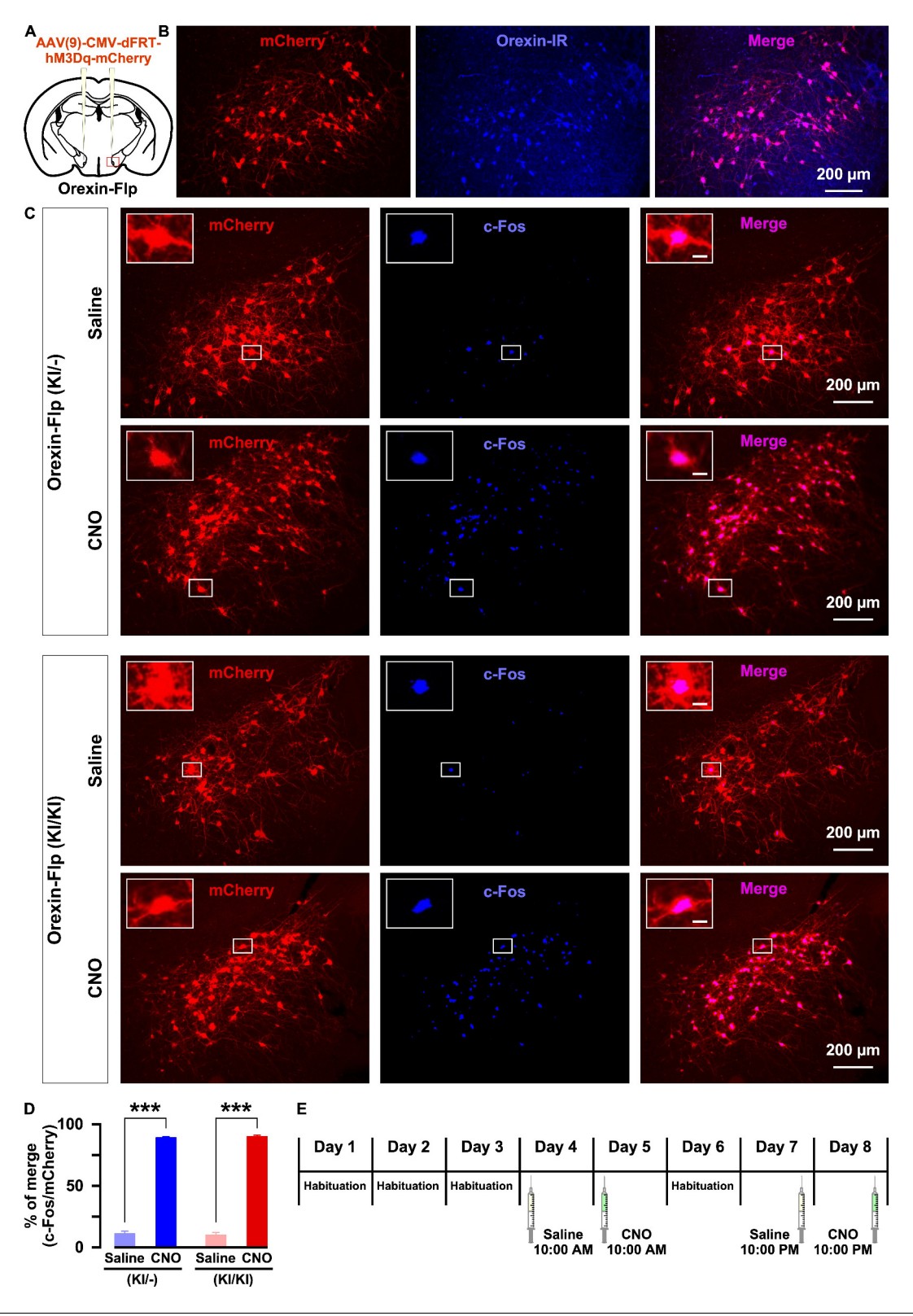

**Figure 6.** Selective chemogenetic activation of orexin neurons. (**A**) Intracranial injection of AAV in OF mice to achieve Flp-dependent expression of hM3Dq-mCherry fusion protein in orexin neurons. (**B**) Representative images showing the expression of mCherry in orexin-immunoreactive neurons in heterozygous OF (KI/-) mice. (**C**) c-Fos expression after i.p. administration of either saline or CNO in both OF (KI/-) and OF (KI/KI) mice. (**D**) Summary of the immunostaining data shown in C. CNO administration can significantly increase neuronal activity in both heterozygous and homozygous mice (n = 3

*Figure 6 continued on next page*

*Figure 6 continued*

mice per group). (E) Schematic showing the schedule of i.p. administration during sleep recording. The p values were determined by a two-tailed student's *t*-test; data represent the mean ± SEM.

DOI: https://doi.org/10.7554/eLife.44927.014

The following source data is available for figure 6:

**Source data 1.** Source data for *Figure 6D*.

DOI: https://doi.org/10.7554/eLife.44927.015

was 11.6 ± 1.9% or 89.2 ± 1.4%, respectively (n = 3 mice/group, p=9.2e-5, *Figure 6C–6D*). In homozygous mice, the ratio of c-Fos expression in mCherry-positive cells of saline or CNO injected mice was 10.6 ± 1.8% or 90.3 ± 1.1%, respectively (n = 3 mice/group, p=9.6e-4, paired *t*-test, *Figure 6C–6D*). Thus, we concluded that the DREADD system successfully enabled selective activation of orexin neurons in both heterozygous and homozygous mice.

## Orexin, and not other co-transmitters, was critical to promote sustained wakefulness and preventing cataplexy

Next, we compared the effect of chemogenetic activation of orexin neurons on sleep/wakefulness. Mice were injected with either saline or CNO during the light (L) period (at 10:00 AM) and the dark (D) period (at 10:00 PM; *Figure 6E*). Expectedly in OF (KI/-) mice, activation of orexin neurons increased total time spent in wakefulness (n = 9 mice, *Figures 7B–I* and *8D–I*) and decreased time in REM sleep (*Figure 7B–ii and D-ii*) and in NREM sleep (*Figure 7B–iii and D-iii*) after CNO administration during both the light and dark periods (*Figure 7—figure supplement 1*). However, while the effects of orexin neuronal activation in OF (KI/KI) mice were comparable to the OF (KI/-) control during the dark period, it was dampened during the light period (n = 8 mice, Wakefulness: *Figure 7B–i and D–I*; REM sleep: *Figure 7B–ii and D-ii*; and NREM sleep: *Figure 7B–iii and B-iii*, and *Figure 7—figure supplement 1*).

This clear difference could be explained by the ability of co-transmitters to partially compensate for the increased wakefulness. Therefore, we hypothesized that the co-transmitters could eventually rescue OF (KI/KI) mice from cataplexy as well. Surprisingly, however, activation of orexin neurons that lacked orexin peptide rather deteriorate the cataplexy during the light periods while it showed similar propensity during the dark period (saline (L): 0.2 ± 0.1 min/hr; CNO (L): 0.9 ± 0.3 min/hr, n = 8 mice, p=0.04; saline (D): 3.0 ± 1.2 min/hr; CNO (D): 5.2 ± 1.4 min/hr, n = 8 mice, p=0.14; *Figure 7B-iv and 7D-iv*). CNO itself is reported to have no effect on the initiation or maintenance of cataplectic attack in the Prepro-orexin KO (OXKO) mice (*Mahoney et al., 2017*). These results clearly suggested that neurotransmitters other than orexin could partially compensate for abnormalities in the regulation of sleep/wakefulness, excluding cataplexy.

We also compared the effect of orexin neuronal activation in the number of episodes (bouts) and average time spent in each vigilance state. Whereas, in OF (KI/-) mice, total observed bouts decrease in all three vigilance states for 2 hr after CNO injection (Wakefulness: *Figure 7C–i and E–i*; REM: *Figure 7C–ii and E-ii*, NREM: *Figure 7C–iii and 7E-iii*), they remain unaffected in case of OF (KI/KI) mice (Wakefulness: *Figure 7C–i and E–i*; REM: *Figure 7C–ii and 7E-ii*, NREM: *Figure 7C–iii and 7E-iii*; Cataplexy: *Figure 7C-iv and 7E-iv*). Conversely, orexin neuronal activation increased average wake duration in OF (KI/-) mice (*Figure 7C–i and E–i*); and decreased REM sleep (*Figure 7C–ii and E-ii*) and NREM sleep duration (*Figure 7C–iii and 7E-iii*). However, in case of OF (KI/KI) mice, while the total wake time was increased after chemogenetic activation (*Figure 7B and 7E i-iv*), duration of wake was not extended (Wakefulness: *Figure 7C–i and E–i*; REM: *Figure 7C–ii and 7E-ii*; NREM: *Figure 7C–iii and 7E-iii*; Cataplexy: *Figure 7C-iv and 7E-iv*). Together, all these experimental analyses indicated that orexin neuropeptide was indispensable to promote sustained arousal, where the co-transmitters could not compensate.

## CNO-induced cataplexy was not different from naïve cataplexy

Finally, we sought to answer the question of whether CNO-induced increase in cataplexy attack showed similar properties to naïve cataplexy. We analyzed and compared the relative power spectrum EEG during the cataplexy episode in the daytime after saline or CNO administration as

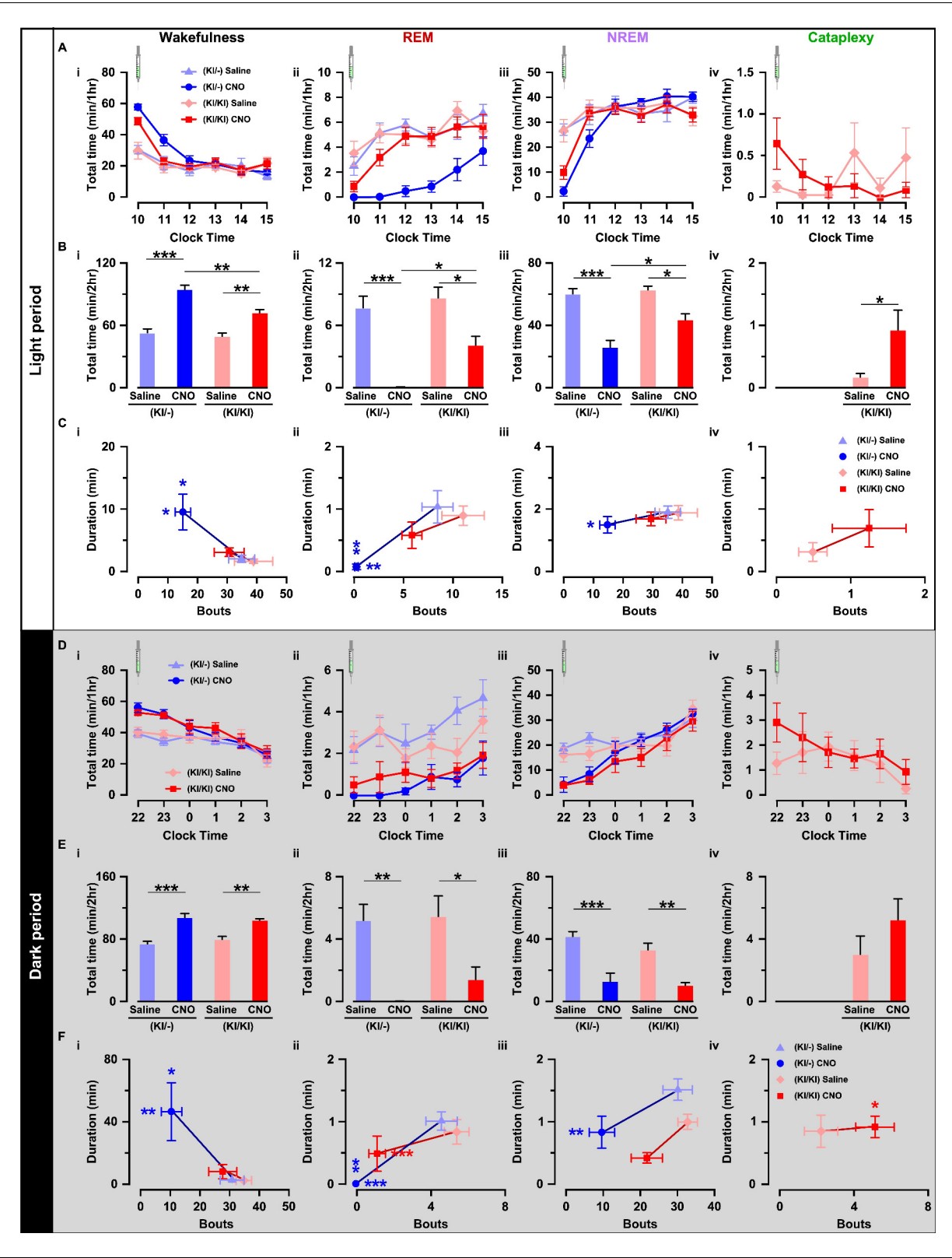

**Figure 7.** Chemogenetic activation of orexin neurons resulted in altered sleep/wakefulness based on the availability of orexin. (**A**) Line graph with symbols showing the time spent in wakefulness (i), REM sleep (ii), NREM sleep (iii), and cataplexy (iv) during each hour for the 6 hr following CNO or saline administration during light period. B, Bar graph showing the 2 hr average of the total time spent in wakefulness (i), REM sleep (ii), NREM sleep (iii), and cataplexy (iv) following CNO or saline injection during the light period. C, Scatter plot showing the averaged bout and duration in wakefulness

*Figure 7 continued on next page*

*Figure 7 continued*

(i), REM sleep (ii), NREM sleep (iii), and cataplexy (iv). The data in D-F are shown similar to the representation in A-C, respectively, recorded during the dark period (OF (KI/-): n = 9 mice and OF (KI/KI): n = 8 mice). Data represent the mean ± SEM in both the line and bar graph. The p values were determined by either two-way ANOVA followed by a post-hoc Bonferroni test or paired student's *t*-test (cataplexy).

DOI: https://doi.org/10.7554/eLife.44927.016

The following source data and figure supplement are available for figure 7:

**Source data 1.** Source data for *Figure 7A, B, C, E and F*.

DOI: https://doi.org/10.7554/eLife.44927.018

**Figure supplement 1.** Representative hypnogram of OF mice after i.p. administration of either saline or CNO.

DOI: https://doi.org/10.7554/eLife.44927.017

chemogenetic activation of orexin KO neurons resulted in increased cataplexy in the daytime (*Figure 8*). Relative power analysis showed no difference in either power of the delta, theta, alpha or beta wave (n = 8 mice, *Figure 8C*), suggesting that CNO-induced cataplexy showed similar electro-cortical activity as the naïve cataplexy attack observed in OF (KI/-) mice.

## Discussion

It has been shown that orexin neurons possess several neurotransmitters and release them together with orexin. However, orexin- or OX2R-KO mice closely phenocopy the symptoms observed in human narcolepsy, which is caused by the specific degeneration of orexin neurons. This might suggest that other neurotransmitters have either no or insignificant role in the regulation of sleep/wakefulness. The physiological importance of orexin can be explored in several ways including central administration, KO, functional manipulation or ablation of source neurons, among others. Here, we dissociated the effect of orexin by applying electrophysiological analyses and neural activity manipulation to orexin neurons that lack orexin peptide using novel KI mice. Previously, we generated Orexin-Cre or Orexin-tTA transgenic mice in which a short 3.2 kb fragment from the 5′-upstream region of the human *hypocretin* gene was used as a promoter (*Inutsuka et al., 2014*; *Sakurai et al., 1999*; *Tabuchi et al., 2013*; *Yamanaka et al., 2003*). However, the random integration of the Orexin-Cre or Orexin-tTA transgene into the genome could result in ectopic gene expression in non-targeted cells as well as unexpected expression during development, as the timing and regulation of gene expression might be affected by genes located near the integrated locus. This could be problematic if these mouse lines were bred with reporter mice. In contrast, KI permits accurate temporo-spatial gene expression control. However, the knock-in method has the disadvantage that orexin content in OF (KI/-) mice could be slightly decreased since Flp was knocked into the *hypocretin* locus. Chemelli *et al.* reported that orexin A content was decreased to 75% in heterozygous mice. (*Chemelli et al., 1999*). OF (KI/-) mice showed a shorter duration of wakefulness compared with WT control mice, and is likely due to the decrease in orexin. The Flp function was restricted in orexin neurons in the OF mice as confirmed by reporter mice. To our knowledge, OF mice represent the first line which enabled exclusive gene expression in orexin neurons by crossing with reporter mice. Moreover, combining Flp and Cre-driver mice enable us to express different genes in different neuronal subtypes to apply neural manipulations and/or activity readout simultaneously. By crossing OF mice with either MCH-Cre or Gad67-Cre, we generated Orexin-Flp; MCH-Cre and Orexin-Flp; Gad67-Cre bigenic mice, and argued that OF mice could be an essential tool for studying the functional connectome between orexin neurons and any other neurons in the hypothalamus. Besides, OF mice can also be useful for analyzing long-range neuronal connections. By using the Orexin-Flp; Gad67-Cre bigenic mice, we recently found that GABAergic neurons in the ventral tegmental area inhibited orexin neuronal activity by making monosynaptic inhibitory projections and regulate NERM sleep in mice (*Chowdhury et al., 2019*).

Previously, we reported that orexin neuropeptide depolarized orexin neurons via OX2R and mediates a positive-feedback loop both directly and indirectly (*Yamanaka et al., 2010*). In agreement with this, we also found that orexin facilitates glutamate release onto orexin neurons. However, several basic electrophysiological properties of orexin neurons were also found to be altered in the absence of orexin peptides, including hyperpolarized membrane potential, lower discharge rate and input resistance. Surprisingly, the capacitance of orexin neurons that lack orexin peptide was

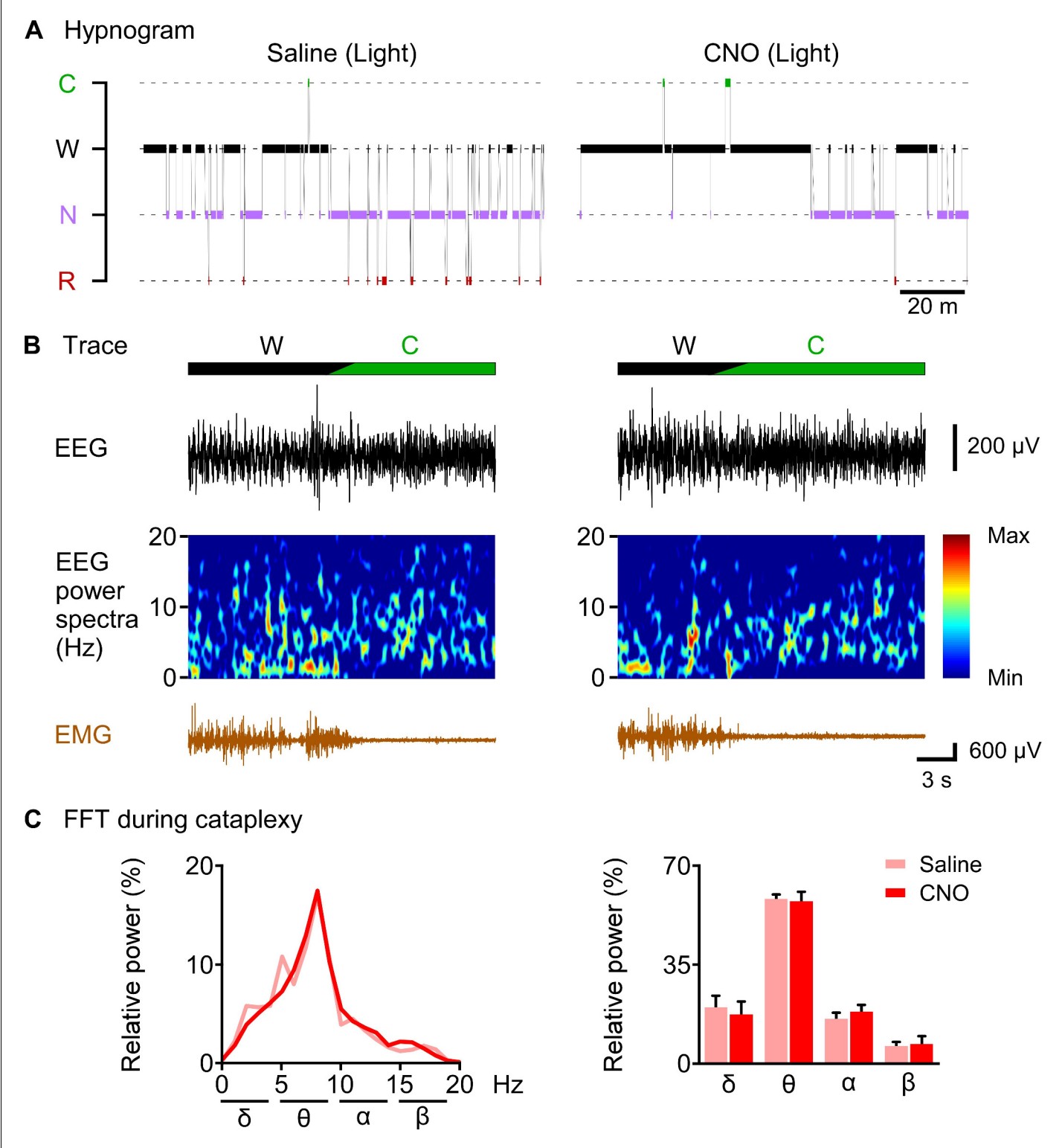

**Figure 8.** CNO-induced cataplexy was not different from naïve cataplexy. (**A**) hypnogram of 2 hr after saline or CNO administration. (**B**) typical traces showing EEG signal, EEG power spectrum and EMG of cataplexy episode. (**C**) line graph (left) and bar graph (right) showing relative power spectrum of EEG during cataplexy episode in the light period (n = 8 mice). Data represent the mean ± SEM in both the line and bar graph.

DOI: https://doi.org/10.7554/eLife.44927.019
The following source data is available for figure 8:

*Figure 8 continued*

**Source data 1.** Source data for *Figure 8C*.

DOI: https://doi.org/10.7554/eLife.44927.020

significantly larger than that of orexin neurons possessing orexin. This indicates that orexin neurons lacking orexin have a larger surface of the cell membrane since capacitance reflects the surface area of the cytoplasmic membrane. We did not directly measure the difference in diameter of cell size since an approximately 3 pF increase is estimated to be associated with a 1–2 µm increase in diameter as a spherical body. This might suggest that orexin signaling through either OX2R or glutamatergic excitatory inputs is involved in the regulation of cell size. However, this study cannot conclude whether the observed altered electrophysiological properties in the orexin KO neurons is directly caused by the absence of orexin, or if the lack of orexin signaling results in developmental compensation that contributes to these differences. Several studies have shown the role of orexin in the developing brain. Satoyanova *et al.* showed that orexin A and B are expressed at 1 week of age during early postnatal brain development (*Stoyanova et al., 2010*). In addition to this, van den Pol *et al.* showed that orexin increased the temporal synchrony of action potentials in developing locus coeruleus cells (*van den Pol et al., 2002*). These reports suggest that orexin might have a role in brain development. To fully characterize some of these unexpected effects of orexin in maintaining the biophysical properties of orexin neurons, in vivo genome editing using the CRISPR/Cas9 system or *hypocretin* gene knockout after maturation using an inducible Cre recombinase such as CreERT2, will likely be required.

Orexin neurons are known to play an essential role in maintaining uninterrupted wakefulness as human narcolepsy patients show chronic daytime sleepiness (*Crocker et al., 2005*; *Thannickal et al., 2000*). Disruption of orexin signaling in mice, rats and dogs produce a very similar phenotype which includes short bouts of wakefulness and increased transitions of vigilance states (*Chemelli et al., 1999*; *Mochizuki et al., 2004*). These suggest that orexin might be the key component released from the orexin neurons in performing the wake-maintaining role of orexin neurons. Our study showed further evidence in support of this hypothesis in the mouse model as a putative increase in the probability of co-transmitters release from orexin neurons could not rescue from the short bout and frequent state-transition phenomenon. Chemogenetic activation of orexin neurons prolonged the duration of wakefulness only in the presence of orexin, suggesting that orexin is crucial to achieve long-duration wakefulness. Moreover, orexin administration can also make wakefulness with high cognitive ability in non-human primates (*Deadwyler et al., 2007*). Such effect of orexin is also expected in human society, and our results indicate the irreplaceability of orexin.

On the contrary, we found that other behavioral outcomes of activating orexin neurons, including increased wakefulness and decreased REM and NREM sleep, can be compensated for by co-transmitters. This effect was obvious especially during the dark (active) period when baseline activity of orexin neurons is presumably higher (*Estabrooke et al., 2001*; *Lee et al., 2005*; *Mileykovskiy et al., 2005*). Interestingly, some other behavioral effects, like the prevention of cataplexy, cannot be compensated for by co-transmitters. Rather, it deteriorates the condition. This might suggest that the role of orexin in the prevention of cataplexy is not simply to activate post-synaptic neurons since glutamate (co-transmitter) also activates post-synaptic neurons. Rather, neurons that are not directly innervated by orexin neurons might also be involved in the regulation of cataplexy. It is also possible that the released orexin diffuses in the CSF since intracerebroventricular injection of orexin inhibited cataplexy (*Mieda et al., 2004*). Moreover, orexin co-transmitter dynorphin, which inhibits orexin neurons both directly and indirectly by depressing glutamatergic afferent inputs to orexin neurons, might also play a key role in such deterioration of cataplexy (*Li and van den Pol, 2006*).

The release of orexin follows a circadian rhythm that is also strongly related to locomotion (*Zhang et al., 2004*). In rodents, the level of orexin peptide in the CSF is low during the light period and high during the active-wake period (*Zeitzer et al., 2003*). Moreover, orexin neurons supposedly show higher activity during active wakefulness and become less active or inactive during REM or NREM sleep (*Lee et al., 2005*; *Mileykovskiy et al., 2005*). Our data also suggest that the activity of orexin neurons is differentially regulated during the light and dark periods. It is possible that the

higher endogenous activity of orexin neurons during the dark period may facilitate CNO-induced transmitter release resulting in improved wakefulness in OF(KI/KI) mice.

In summary, here we dissociated the role of orexin at the cellular and behavioral level. We suggest that the primary function of orexin is to maintain the electrophysiological balance, the input-output functions in orexin neurons and most importantly, to exert the function of orexin neurons in maintaining sustained wakefulness.

# Materials and methods

## Key resources table

| Reagent type (species) or resource | Designation | Source or reference | Identifiers | Additional information |
|---|---|---|---|---|
| Strain, strain background | AAV(9)-CMV-dFRT-hM3Dq-mCherry | This paper | NA | Titer: $1.0 \times 10^{12}$ particles/ml |
| Strain, strain background | AAV(DJ)-CMV-dFRT-hrGFP | This paper | NA | Titer: $3.7 \times 10^{12}$ particles/ml |
| Strain, strain background | AAV(DJ)-CMV-FLEX-tdTomato | This paper | NA | Titer: $2.0 \times 10^{12}$ particles/ml |
| Genetic reagent (*Mus musculus*) | Orexin-Flippase | This paper | Orexin-Flp | Knock-in mice; EGFP-2A-Flp was placed just behind the translational initiation site of the *hypocretin* gene in-frame |
| Genetic reagent (*Mus musculus*) | Glutamic acid decarboxylase 67-Cre | *Higo et al., 2009* | Gad67-Cre | |
| Genetic reagent (*Mus musculus*) | R26-CAG-FRT-STOP-FRT-mTFP1 | *Imayoshi et al., 2012* | FSF-mTFP1 | |
| Genetic reagent (*Mus musculus*) | Melanin-concentrating hormone-Cre | *Kong et al., 2010* | MCH-Cre | |
| Genetic reagent (*Mus musculus*) | Orexin-enhanced green fluorescence protein | *Yamanaka et al., 2003* | Orexin-EGFP | |
| Antibody | goat polyclonal anti-orexin | Santa Cruz Biotechnology | sc-8070 | (1/1000) |
| Antibody | rabbit polyclonal anti-MCH | Sigma-Aldrich | M8440 | (1/2000) |
| Antibody | rabbit polyclonal anti-prodynorphin | Merck Millipore | AB5519 | (1/100) |
| Antibody | mouse monoclonal anti-c-Fos | Santa Cruz Biotechnology | E-8 | (1/500) |
| Antibody | mouse monoclonal anti-DsRED | Santa Cruz Biotechnology | sc-390909 | (1/1000) |
| Antibody | mouse monoclonal anti-GFP | Fujifilm Wako Pure Chemical Corporation | mFX75 | (1/1000) |
| Sequence-based reagent (primer) | forward primer: 5'-CTCATTAGTAC TCGGAAACTGCCC-3' | This paper | NA | Primer for PCR genotyping of orexin-Flp mouse tail DNA |
| Sequence-based reagent (primer) | reverse primer: 5'-AAGCACTATCATG GCCTCAGTAGT-3' | This paper | NA | Primer for PCR genotyping of orexin-Flp mouse tail DNA |
| Chemical compound, drug | clozapine-N-oxide (CNO) | Enzo Life Sciences | BML-NS105-0025 | |
| Chemical compound, drug | Picrotoxin | Sigma-Aldrich | P1675 | |

*Continued on next page*

*Continued*

| Reagent type (species) or resource | Designation | Source or reference | Identifiers | Additional information |
|---|---|---|---|---|
| Chemical compound, drug | tetrodotoxin (TTX) | Alomone Labs | Cat# T-550 | |
| Chemical compound, drug | D-2-Amino-5-phosphopentanoic acid (AP5) | Alomone Labs | Cat# D-145 | |
| Chemical compound, drug | 6-cyano-7-nitroquinoxaline-2,3-dione (CNQX) | Sigma-Aldrich | Cat# C127 | |
| Chemical compound, drug | Lidocaine N-ethyl bromide (QX-314 bromide) | Alomone Labs | Cat# Q-100 | |
| Software, algorithm | Origin 2017 | Lightstone | Origin 2018 | |
| Software, algorithm | SleepSign | Kissei Comtec | Version 3 | |
| Software, algorithm | pClamp 10.5Software and Algorithms | Molecular Devices | RRID:SCR_011323 | |
| Software, algorithm | ImageJ | https://imagej.nih.gov/ij/ | RRID:SCR_003070 | |
| Other | DAPI stain | Thermo Fisher Scientific | Cat# D1306 | |

## Subjects

All experimental protocols in this study involving the use of mice were approved and were performed in accordance with the approved guidelines of the Institutional Animal Care and Use Committees of the Research Institute of Environmental Medicine, Nagoya University, Japan (Approval number #18232, #18239). Mice were group housed unless stated otherwise, on a 12 hr light-dark cycle (lights were turned on at 8:00 AM), with free access to food and water. All efforts were made to reduce the number of animals used and to minimize the suffering and pain of animals.

## Generation of OF knock-in mice

To generate the OF knock-in mice, we designed a targeting vector in which Flp recombinase cDNA was fused to enhanced green fluorescent protein (EGFP) with the 2A peptide gene (EGFP-2A-Flp) and was placed just behind the translational initiation site of the *hypocretin* gene in-frame. The knock-in vector was constructed with the MC1 promoter-driven diphtheria toxin gene, a 5.44 kb fragment at the 5' site. Flp recombinase, including a nuclear localization signal cDNA, was fused to EGFP with the T2A peptide sequence, a *Pgk-1* promoter-driven neomycin phosphotransferase gene (neo) flanked by two Dre recognition target (rox) sites, and a 5.16 kb fragment at the 3' site (*Figure 1A*). The sequence of EGFP-2A-Flp was codon-optimized for expression in mammalian cells. Linearized targeting vector was electroporated into embryonic stem cells from the C57BL/6 mouse line (RENKA), and corrected targeted clones were isolated by southern blotting. Two founders were obtained, and the B line was used in this study. PCR genotyping of mouse tail DNA was performed with the following primers: knock-in forward, 5'-CTCATTAGTACTCGGAAACTGCCC-3'; knock-in reverse, 5'-AAGCACTATCATGGCCTCAGTAGT-3'.

## Generation of Orexin-Flp; FSF-mTFP1, Orexin-Flp; MCH-Cre and Orexin-Flp; Gad67-Cre bigenic mice

After several generations of breeding, OF mice were separately bred with either R26-CAG-FRT-STOP-FRT-mTFP1 (FSF-mTFP1) (*Imayoshi et al., 2012*), MCH-Cre (*Kong et al., 2010*), or glutamic acid decarboxylase at 67 K-dalton (Gad67)-Cre (Gad67-Cre) (*Higo et al., 2009*) mice to generate Orexin-Flp; FSF-mTFP1, Orexin-Flp; MCH-Cre or Orexin-Flp; Gad67-Cre bigenic mice, respectively.

## Generation and microinjection of viral vectors

Adeno-associated viral (AAV) vectors were produced using the AAV Helper-Free System (Agilent Technologies, Inc, Santa Clara, CA, USA). The virus purification method was modified from a previously published protocol (*Inutsuka et al., 2016*). Briefly, HEK293 cells were transfected with a pAAV vector, pHelper and pAAV-RC (serotype nine or DJ; purchased from Cell Biolabs Inc, San Diego, CA, USA) plasmid using a standard calcium phosphate method. Three days after transfection, cells were collected and suspended in artificial CSF (aCSF) solution (in mM: 124 NaCl, 3 KCl, 26 NaHCO$_3$, 2 CaCl$_2$, 1 MgSO$_4$, 1.25 KH$_2$PO$_4$ and 10 glucose). Following multiple freeze-thaw cycles, the cell lysates were treated with benzonase nuclease (Merck, Darmstadt, Germany) at 37°C for 30 min, and were centrifuged 2 times at 16,000 g for 10 min at 4°C. The supernatant was used as the virus-containing solution. Quantitative PCR was performed to measure the titer of purified virus. Virus aliquots were then stored at −80°C until use.

A dFRT cassette was used for Flp-dependent gene expression control. A dFRT cassette is composed of two different FRT sequences (FRT and F3) located in a cis position. In the presence of Flp, sequence between dFRT is reversed and fixed. To express hM3Dq exclusively in Flp-expressing neurons, we stereotactically injected 600 nl of AAV(9)-CMV-dFRT-hM3Dq-mCherry (viral titer: $1.0 \times 10^{12}$ particles/ml) virus into both brain hemispheres of OF mice using the following coordinates: −1.4 mm posterior to the bregma, 0.8 mm lateral to the midline, −5.0 mm ventral to the brain surface. Mice were anesthetized with 1.5–2.0% isoflurane (Wako Pure Chemical Industries, Osaka, Japan) using a Univentor 400 Anaesthesia Unit (Univentor Ltd., Malta) throughout the entire surgery. These mice were used in the behavioral experiments beginning at least 3 weeks post-injection. For Flp-dependent expression of humanized Renilla reniformis green fluorescent protein (hrGFP) and Cre-dependent expression of tdTomato, we stereotactically injected 600 nl of virus cocktail containing equal volume of AAV(DJ)-CMV-dFRT-hrGFP (viral titer: $3.7 \times 10^{12}$ particles/ml) and AAV(DJ)-CMV-FLEX-tdTomato (viral titer: $2.0 \times 10^{12}$ particles/ml) into one brain hemisphere of OF mice using the same coordinates described above.

## Immunohistochemistry

Mice were anesthetized with 10% somnopentyl (1.0 mg/kg, Kyoritsu Seiyaku Corporation, Tokyo, Japan) and were perfused transcardially with 20 ml of ice-cold saline. This perfusion was immediately followed by another 20 ml of 10% ice-cold formalin (Wako Pure Chemical Industries, Ltd., Osaka, Japan). Brains were then isolated and postfixed in 10% formalin solution at 4°C overnight. Subsequently, brains were immersed in 30% sucrose in PBS at 4°C for at least 2 days. Coronal brain slices of 40 μm thickness were generated using a cryostat (Leica CM3050 S; Leica Microsystems, Wetzlar, Germany). For staining, coronal brain sections were immersed in blocking buffer (1% BSA and 0.25% Triton-X in PBS), then incubated with primary antibodies at 4°C overnight. The sections were then washed with blocking buffer and incubated with secondary antibodies for 1 hr at room temperature (RT). After washing, brain sections were mounted and examined using a fluorescence microscope (BZ-9000, Keyence, Osaka, Japan or IX71, Olympus, Tokyo, Japan).

## Antibodies and stains

Primary antibodies were diluted in the blocking buffer as follows: anti-orexin-A goat antibody (Santa Cruz, Dallas, TX) at 1:1000, anti-MCH rabbit antibody (Sigma-Aldrich) at 1:2000, anti-prodynorphin guinea pig antibody (Merck Millipore, Billerica, MA) at 1:100, anti-c-Fos rabbit antibody (Santa Cruz) at 1:500 and anti-GFP mouse antibody (Wako, Japan) at 1:1000. Secondary antibodies included: CF 488- or CF 594-conjugated anti-goat antibody (Biotium Inc, Hayward, CA), CF 647-conjugated anti-rabbit antibody (Biotium), CF 680-conjugated anti-guinea pig antibody (Biotium) and CF 488-conjugated anti-mouse antibody (Biotium); all were diluted at 1:1000 in blocking buffer.

## Acute brain slice preparations and electrophysiological recording

OF and OE (*Yamanaka et al., 2003*) mice of both sexes, aged 2–5 months, were used for electrophysiological recordings. Brain slice preparations and subsequent electrophysiological recording were modified from a previously published protocol (*Chowdhury and Yamanaka, 2016*). Briefly, mice were deeply anesthetized using isoflurane (Wako) and decapitated at around 11:00 AM. Brains were quickly isolated and chilled in ice-cold oxygenated (95% O$_2$ and 5% CO$_2$) cutting solution (in

mM: 110 K-gluconate, 15 KCl, 0.05 EGTA, 5 HEPES, 26.2 NaHCO$_3$, 25 Glucose, 3.3 MgCl$_2$ and 0.0015 ($\pm$)$-$3-(2-Carboxypiperazin-4-yl)propyl-1-phosphonic acid). After trimming the brain, coronal brain slices of 300 µm thickness that contained the LHA were generated using a vibratome (VT-1200S; Leica, Wetzlar, Germany) and were temporarily placed in an incubation chamber containing oxygenated bath solution (in mM: 124 NaCl, 3 KCl, 2 MgCl$_2$, 2 CaCl$_2$, 1.23 NaH$_2$PO$_4$, 26 NaHCO$_3$ and 25 Glucose) in a 35°C water bath for 60 min. Slices were then incubated at RT in the same incubation chamber for another 30–60 min for recovery.

Acute brain slices were transferred from the incubation chamber to a recording chamber (RC-26G; Warner Instruments, Hamden, CT, USA) equipped with an upright fluorescence microscope (BX51WI; Olympus, Tokyo, Japan), and were superfused with oxygenated bath solution at the rate of 1.5 ml/min using a peristaltic pump (Dynamax; Rainin, Oakland, CA, USA). An infrared camera (C3077-78; Hamamatsu Photonics, Hamamatsu, Japan) was installed in the fluorescence microscope along with an electron multiplying charge-coupled device camera (EMCCD, Evolve 512 delta; Photometrics, Tucson, AZ, USA) and both images were separately displayed on monitors. Micropipettes of 4–6 MΩ resistance were prepared from borosilicate glass capillaries (GC150-10; Harvard Apparatus, Cambridge, MA, USA) using a horizontal puller (P-1000; Sutter Instrument, Novato, CA, USA). Patch pipettes were filled with KCl-based internal solution (in mM: 145 KCl, 1 MgCl$_2$, 10 HEPES, 1.1 EGTA, 2 MgATP, 0.5 Na$_2$GTP; pH 7.3 with KOH) with osmolality between 280–290 mOsm. Positive pressure was introduced in the patch pipette as it approached the cell. For whole-cell current clamp or voltage clamp recordings, a giga-seal of resistance >1 GΩ was made between the patch pipette and the cell membrane by releasing the positive pressure upon contacting the cell. The patch membrane was then ruptured by gentle suction to form a whole-cell configuration. Electrophysiological properties of the cells were monitored using the Axopatch 200B amplifier (Axon Instrument, Molecular Devices, Sunnyvale, CA). Output signals were low-pass filtered at 5 kHz and digitized at a 10 kHz sampling rate. Patch clamp data were recorded through an analog-to-digital (AD) converter (Digidata 1550A; Molecular Devices) using pClamp 10.2 software (Molecular Devices). Blue light with a wavelength of 475 ± 17.5 nm was generated by a light source that used a light-emitting diode (Spectra light engine; Lumencor, Beaverton, OR, USA) and was guided to the microscope stage with a 1.0 cm diameter fiber. Brain slices were illuminated through the objective lens of the fluorescence microscope. EGFP-expressing orexin neurons were identified by its fluorescence. For cell-attached recording, a seal of resistance <1 GΩ was made and spontaneous firing was recorded. The resting membrane potentials (Vrest) were measured from offline analysis of current clamp recordings using the predefined fitting function provided by Clampfit. We performed standard exponential fitting with zero shift for the initial 20 s of data to measure the Vrest of recorded neurons. Firing frequency was also calculated from offline analysis of the initial 60 s of the cell-attached recording data. sEPSCs were recorded with picrotoxin (400 µM) and sIPSCs were recorded with AP-5 (50 µM) and CNQX (20 µM) in bath solutions. Both sEPSCs and sIPSCs were recorded in the presence of KCl-based pipette solutions that included 1 mM of QX-314.

## EEG-EMG surgery, data acquisition, and vigilance state determination

Virus injected age-matched male mice were implanted with EEG and EMG electrodes for polysomnographic recording under isoflurane anesthesia following the protocol published elsewhere (*Tabuchi et al., 2014*). Immediately after surgery, each mouse received an i.p. injection of 10 ml/kg of analgesic solution containing 0.5 mg/ml of Carprofen (Zoetis Inc, Japan). The same analgesic at the same dose was administered again 1 day after surgery. Mice were singly housed for 7 days during recovery. Mice were then connected to a cable with a slip ring in order to move freely in the cage and were habituated with the cable for another 7 days. The first 3 days were treated as the adaptation period for the animals to acclimate to the new environment and to intraperitoneal (i.p.) administration (10 ml/kg) of saline. On days 4 and 5, the mice were injected with saline (day 4) and CNO (Enzo Life Sciences, Farmingdale, NY, USA) (day 5) at 10:00 AM during the light period. On days 7 and 8, they were injected with saline (day 7) and CNO (day 8) at 10:00 PM during the dark period (See *Figure 6*). CNO was dissolved in water to make a stock solution (10 mg/ml) and was diluted with saline to a final concentration of 100 µg/ml just prior to i.p. administration.

EEG and EMG signals were amplified (AB-610J, Nihon Koden, Japan), filtered (EEG 1.5–30 Hz and EMG 15–300 Hz), digitized (at a sampling rate of 128 Hz), recorded (Vital Recorder, Kissei Comtec Co., Ltd, Japan) and finally analyzed (SleepSign, Kissei Comtec). Animal behavior was monitored

through a CCD video camera (Amaki Electric Co., Ltd., Japan) during both the light and dark periods. The dark period video recording was assisted by infrared photography (Amaki Electric Co., Ltd., Japan) and an infrared sensor (Kissei Comtec). EEG and EMG data were automatically scored in 4 s epochs and classified as wake, rapid eye movement sleep, and non-rapid eye movement sleep. The EEG analysis yielded power spectra profiles over a 0 ~ 20 Hz window with 1 Hz resolution for delta (1–5 Hz), theta (6–10 Hz), alpha (11–15 Hz), and beta (16–20 Hz) bandwidths. All auto-screened data were examined visually and corrected. The criteria for vigilance states were the same as described previously (*Tabuchi et al., 2014*). Briefly, (i) wake (low EEG amplitude with high EMG or locomotion score), (ii) non-rapid eye movement (NREM) sleep (low EMG and high EEG delta amplitude), and (iii) rapid eye movement (REM) sleep (low EMG, low EEG amplitude with high theta activity, and should be followed by NREM). Cataplexy was tracked using a combination of multiple criteria: muscle atonia lasting $\geq 10$ s, predominance of theta activity and more than 40 s of wakefulness before the cataplectic attack.

## Data analysis and presentation

Immunostaining data were analyzed and processed with ImageJ (US National Institute of Health) and BZ-X Analyzer (Keyence BZ-X710 microscope). Electrophysiological analysis was performed with either Clampfit10 (Molecular Devices, Sunnyvale, CA) or Minianalysis software (Synaptosoft Inc, Decatur, GA). Electrophysiological data were saved as American Standard Code for Information Interchange (ASCII) files and further data calculations were performed in Microsoft Excel. Graphs were generated in Origin 2017 (OriginLab, Northampton, MA) using data from Excel. Statistical analysis was also performed with Origin 2017. Graphs were generated using Canvas 15 (ACD Systems, Seattle, WA).

## Experimental design and statistical analysis

The electrophysiological effects of knocking out a single neuropeptide in its source neurons were analyzed by slice electrophysiology. Individual sample sizes for slice patch-clamp recordings (n number of neurons) are reported separately for each experiment. The physiological effects of activating orexin neurons that lack orexin neuropeptide were also analyzed. In all cases, five or more animals were used for each parameter tested. All statistical tests, including the exact p values, are described when used. No statistical analyses were used to predetermine sample sizes. All data are presented as the mean ± standard error of the mean (SEM). For all statistical tests *p<0.05, **p<0.01, ***p<0.001 were considered significant and p>0.05 was considered not significant (ns).

## Acknowledgements

We thank S Tsukamoto, Y Miyoshi, A Inui, and G Wang for technical assistance.

## Additional information

### Funding

| Funder | Grant reference number | Author |
| --- | --- | --- |
| Japan Science and Technology Agency | JPMJCR1656 | Akihiro Yamanaka |
| Ministry of Education, Culture, Sports, Science, and Technology | 26293046 | Akihiro Yamanaka |
| Ministry of Education, Culture, Sports, Science and Technology | 26640041 | Akihiro Yamanaka |
| Ministry of Education, Culture, Sports, Science and Technology | 16H01271 | Akihiro Yamanaka |

| Ministry of Education, Culture, Sports, Science and Technology | 15H01428 | Akihiro Yamanaka |
| Ministry of Education, Culture, Sports, Science and Technology | 18KK0223 | Akihiro Yamanaka |
| Ministry of Education, Culture, Sports, Science and Technology | 19H05016 | Akihiro Yamanaka |

The funders had no role in study design, data collection and interpretation, or the decision to submit the work for publication.

### Author contributions
Srikanta Chowdhury, Conceptualization, Data curation, Formal analysis, Writing—original draft, Writing—review and editing; Chi Jung Hung, Data curation, Formal analysis, Investigation; Shuntaro Izawa, Data curation; Ayumu Inutsuka, Conceptualization, Resources; Meiko Kawamura, Takashi Kawashima, Haruhiko Bito, Itaru Imayoshi, Manabu Abe, Kenji Sakimura, Resources; Akihiro Yamanaka, Conceptualization, Supervision, Funding acquisition, Writing—original draft, Writing—review and editing

### Author ORCIDs
Srikanta Chowdhury (iD) https://orcid.org/0000-0002-2216-5960
Chi Jung Hung (iD) https://orcid.org/0000-0002-1169-7953
Ayumu Inutsuka (iD) http://orcid.org/0000-0001-6503-5501
Akihiro Yamanaka (iD) https://orcid.org/0000-0001-6099-7306

### Ethics
Animal experimentation: All experimental protocols in this study involving the use of mice were approved and were performed in accordance with the approved guidelines of the Institutional Animal Care and Use Committees of the Research Institute of Environmental Medicine, Nagoya University, Japan (Approval number #18232, #18239).

### Decision letter and Author response
Decision letter https://doi.org/10.7554/eLife.44927.025
Author response https://doi.org/10.7554/eLife.44927.026

## Additional files

### Supplementary files
• Transparent reporting form
DOI: https://doi.org/10.7554/eLife.44927.021

### Data availability
All data generated or analysed during this study are included in the manuscript and supporting files.

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
