## [Decision Letter]

Thank you for submitting your article "Dissociating orexin-dependent and -independent functions of orexin neurons using novel *orexin-Flp*knock-in mice" for consideration by *eLife*. Your article has been reviewed by three peer reviewers, including Yang Dan as the Reviewing Editor and Reviewer #1, and the evaluation has been overseen by Catherine Dulac as the Senior Editor.

The reviewers have discussed the reviews with one another and the Reviewing Editor has drafted this decision to help you prepare a revised submission.

Summary:

Orexin neurons are known to be important for sustained arousal, and they release glutamate and dynorphin in addition to orexin. This paper addresses the effect of knocking out orexin on the property of orexinergic neurons and on the outcome of activating orexinergic neurons. They found that orexin knock out altered the electrophysiological properties of the neurons and reduced their glutamatergic inputs. While all transmitters contribute to arousal, orexin is crucial for sustained arousal and prevention of cataplexy. Interestingly, activating orexinergic neurons in an orexin deficient background causes increase in cataplexy. This novel finding is expected to lead to future discoveries on the mechanism of cataplexy. The new mouse line they generated is of great interest to the field, given the highly specific expression of Flp in orexin neurons.

Essential revisions:

Discussion section: "The Flp function was restricted in orexin neurons in the OF mice as confirmed by reporter mice."

To conclude this, it is essential to describe how many% of reporter positive cells are also orexin positive, which seems not provided. (The authors only state how many% of orexin positive cells are also reporter positive.)

Subsection “OF (KI/KI) mice showed symptoms in narcolepsy”: "Sleep state parameters for orexin knockout mice are presented in Table 1. These values were comparable with our previously generated orexin neuron-ablated mice (Tabuchi et al., 2014a), as well as with the previously generated orexin-knockout mice (Chemelli et al., 1999)."

This part is not very convincing. First, in case of Tabuchi et al., 2014a, the amount of each sleep stage changes dynamically following DTA expression so it is difficult to compare sleep amount to the current KI mice. According to Table 1, the KI mice show high REM sleep amount (96 min.), whereas NREM sleep amount is quite low (584 min.). According to Chemelli et al., 1999, Orexin KO mice exhibit normal amount of REM sleep (88 min.) and high amount of NREM sleep (761 min.). Thus, contrary to the manuscript, the values do not seem "comparable" to both papers that were cited. In addition, according to Hara et al., 2001, mice in which orexinergic neurons were ablated by ataxin-expression exhibited a similar trend to Chemelli et al., 1999, i.e. normal amount of REM sleep (88 min.) and high amount of NREM sleep (731 min.). The characterization of the sleep architecture of the KI mice generated here is important because unexpected effects of expressing Flp in these neurons are possible. The recording environment might have affected the sleep differently from Chemelli et al., 1999 or Hara et al., 2001. To truly characterize the KI mice, please also provide sleep data of wt control mice.

The authors find that chemogenetic activation of orexin neurons in the KI/- mice strongly increases the average duration of wake bouts, while activation of orexin neurons w/o orexin in the KI/KI mice has no effect on wake duration (Figure 7C, F). The control recordings (with saline injections) show no significant differences between the two mouse lines. However, also during spontaneous sleep/wake behavior mice show sustained periods of wakefulness, especially during the dark period (Lo et al., 2004, Stephenson et al., 2013), implying that the control recordings with saline in Figure 7 should contain long wake periods. Consequently, if the release of orexin by orexin neurons was indispensable for sustained wake periods, KI/KI mice (w/o orexin in orexin neurons) should show shortened wake periods under control (saline) conditions. As this is not the case, the reported data in their present form demonstrate that orexin is sufficient, but not indispensable for sustained wake periods. This point should be clarified.

As the duration of wake durations typically follows a bimodal distribution with short and long wake bouts (Lo et al., 2004, Stephenson et al., 2013), potential effects on long wake periods in the control recordings might be occluded by averaging. By separating the distribution of wake periods into short and long periods, the authors could test whether the absence of orexin specifically affects either of these two distributions. A second reason why the authors could not detect differences under control conditions might be that the analyses in Figure 7C, F only include the first two hours after saline injection (which might not contain enough wake periods to detect differences). By extending their analysis to the complete control recordings, can the authors find shortened wake periods in KI/KI mice supporting a necessary role of orexin in sustaining wakefulness?

---

## [Author Response]

Essential revisions:Discussion section: "The Flp function was restricted in orexin neurons in the OF mice as confirmed by reporter mice."To conclude this, it is essential to describe how many% of reporter positive cells are also orexin positive, which seems not provided. (The authors only state how many% of orexin positive cells are also reporter positive.)

In the previous version of the manuscript, we stated that “We observed no ectopic expression of EGFP and/or mTFP1 protein in melanin-concentrating hormone (MCH)-immunoreactive (n = 4 mice; Figure 1C and 1D) or in any other non-orexin-immunoreactive neurons in these bigenic mice”. However, to further clarify this description, we added the following sentence:

“In other words, all reporter gene-expressing neurons were orexin-immunoreactive (IR).”

In addition, we added a bar graph to show the percentage of reporter gene-positive cells that were also orexin-positive in Figure 1D.

Subsection “OF (KI/KI) mice showed symptoms in narcolepsy”: "Sleep state parameters for orexin knockout mice are presented in Table 1. These values were comparable with our previously generated orexin neuron-ablated mice (Tabuchi et al., 2014a), as well as with the previously generated orexin-knockout mice (Chemelli et al., 1999)."This part is not very convincing. First, in case of Tabuchi et al., 2014a, the amount of each sleep stage changes dynamically following DTA expression so it is difficult to compare sleep amount to the current KI mice. According to Table 1, the KI mice show high REM sleep amount (96 min.), whereas NREM sleep amount is quite low (584 min.). According to Chemelli et al., 1999, Orexin KO mice exhibit normal amount of REM sleep (88 min.) and high amount of NREM sleep (761 min.). Thus, contrary to the manuscript, the values do not seem "comparable" to both papers that were cited. In addition, according to Hara et al., 2001, mice in which orexinergic neurons were ablated by ataxin-expression exhibited a similar trend to Chemelli et al., 1999, i.e. normal amount of REM sleep (88 min.) and high amount of NREM sleep (731 min.). The characterization of the sleep architecture of the KI mice generated here is important because unexpected effects of expressing Flp in these neurons are possible. The recording environment might have affected the sleep differently from Chemelli et al., 1999 or Hara et al., 2001. To truly characterize the KI mice, please also provide sleep data of wt control mice.

As requested by the reviewer, we performed sleep recordings and analysis in wild type (WT) control mice (n = 8) and compared these results with *OF* (KI/KI) mice (n = 7). These results have been added to Table 1. *OF* (KI/KI) mice showed a significant increase in the number of bouts and a decreased average duration in all vigilance states compared with WT mice. These results suggest fragmentation of sleep/wakefulness and an inability to maintain sustained arousal in the absence of orexin. However, we observed a higher amount of REM sleep even in the WT control mice (92 min) compared with values reported for WT mice by Chemelli et al., 1999 (73 min). Although our REM sleep results for WT mice were similar to that reported by Hara et al., 2001 (92 min), our WT mice showed a higher level of wakefulness (792 min) and less NREM sleep (556 min) than observed by either Chemelli et al. (awake: 661 min, NREM: 703 min) or Hara et al. (awake: 654 min, NREM: 692 min). These differences might be caused by either i) the position of the EEG electrodes, or ii) the criteria used for the definition of vigilance state and the recording environment.

i) Position of EEG electrodes:

Our position: AP -1.5, ML -1.0 and AP -5.4, ML -1.0, mm.

Chemelli et al., 1999: AP1.1, ML ± 1.45 and AP-3.5, ML ± 1.45, mm.

Duran et al. (Sleep 2018) reported that differences in electrode position resulted in different REM sleep times. Our electrode position enabled detection of theta activity from the hippocampus with high sensitivity since the electrode was positioned close to the hippocampus. Thus, the observed increase in REM sleep time under our recording conditions compared with other reports might be caused by such differences in electrode position.

ii) Criteria for the definition of vigilance state: Our definition of drowsiness differed from that used by others. While Chemelli et al. defined drowsiness as NREM sleep, we defined it as wakefulness since we identified NREM sleep as having a low EMG and high delta amplitude by EEG. This might be caused by less NREM sleep and longer wakefulness compared with previous reports.

Therefore, we hypothesize that one potential reason for the differences in the amount of NREM and REM sleep in orexin-Flp mice may be the differences in recording conditions, environment and criteria used for sleep/wakefulness analysis, and not due to the expression of Flp. Moreover, patch clamp recording from orexin neurons in orexin-Flp (KI/-) heterozygous mice showed no abnormality in electrophysiological parameters compare with orexin-EGFP mice, confirming that Flp expression did not induce unexpected effect (Figure 4). Nevertheless, to avoid any confusion, we have removed the citation of Tabuchi et al. and added Hara et al. We added the following sentences to the Results:

“We also compared the vigilance state parameters of the *OF* (KI/KI) mice with those of wild type (WT) control mice and the values are presented in Table 1. As expected, *OF* (KI/KI) mice showed less wakefulness and higher NREM and REM sleep compared to the WT control animals (Table 1). Along with these changes, orexin KO mice showed an increased number of bouts and decreased average duration in all vigilance states (Table 1), suggesting fragmentation of the sleep-wakefulness and an inability to maintain sustained arousal in the absence of orexin peptides. However, in comparison to the previously generated orexin KO (Chemelli et al., 1999) as well as to the orexin neuron-ablated mice (Hara et al., 2001), newly generated *OF* (KI/KI) mice showed higher REM sleep and wakefulness and a lower amount of NREM sleep. This is presumably due to changes in the recording environment, different criteria of the definition of vigilance state and/or changes in the genetic background as the WT control mice show similar changes.”

The authors find that chemogenetic activation of orexin neurons in the KI/- mice strongly increases the average duration of wake bouts, while activation of orexin neurons w/o orexin in the KI/KI mice has no effect on wake duration (Figure 7C, F). The control recordings (with saline injections) show no significant differences between the two mouse lines. However, also during spontaneous sleep/wake behavior mice show sustained periods of wakefulness, especially during the dark period (Lo et al., 2004, Stephenson et al., 2013), implying that the control recordings with saline in Figure 7 should contain long wake periods. Consequently, if the release of orexin by orexin neurons was indispensable for sustained wake periods, KI/KI mice (w/o orexin in orexin neurons) should show shortened wake periods under control (saline) conditions. As this is not the case, the reported data in their present form demonstrate that orexin is sufficient, but not indispensable for sustained wake periods. This point should be clarified.As the duration of wake durations typically follows a bimodal distribution with short and long wake bouts (Lo et al., 2004, Stephenson et al., 2013), potential effects on long wake periods in the control recordings might be occluded by averaging. By separating the distribution of wake periods into short and long periods, the authors could test whether the absence of orexin specifically affects either of these two distributions. A second reason why the authors could not detect differences under control conditions might be that the analyses in Figure 7C,F only include the first two hours after saline injection (which might not contain enough wake periods to detect differences). By extending their analysis to the complete control recordings, can the authors find shortened wake periods in KI/KI mice supporting a necessary role of orexin in sustaining wakefulness?

To address whether KI/KI mice (that lack orexin expression in orexin neurons) exhibit shortened periods of wakefulness under control (saline) conditions, we extended our analysis for 12 hours after the injection of saline. However, the duration and number of bouts of wakefulness were not significantly different. The duration of wakefulness in heterozygous (n = 4) and homozygous (n = 6) mice were 106.3 ± 16.8 sec and 99.5 ± 11.9 sec (Author response image 1 = 0.7, unpaired t-test), respectively. The number of bouts of wakefulness in heterozygous and homozygous mice was 228.0 ± 23.1 and 252.8 ± 20.8 (Author response image 1 = 0.5, unpaired t-test), respectively.

**Author response image 1. respfig1:** Bouts and duration of wakefulness in *OF* (KI/-) (n = 4) and *OF* (KI/KI) (n = 7) mice after saline injection for 12 hr). Data represent mean ± S.E.M.

It is, however, notable that both the duration and bouts of wakefulness in heterozygous mice was not comparable to those of WT mice (Table 1; duration: 316.6 ± 30.9 sec, bouts: 107.5 ± 10.6, without saline injection during the dark period). This is presumably because of decrease in orexin in heterozygous mice since the Flp gene was knocked-in to the prepro-orexin gene locus. We could not compare sleep/wakefulness values from heterozygous mice with previous reports including that of Chemelli et al., since they did not provide such values. However, Chemelli et al. did describe a decrease in the content of orexin peptides in the brain of heterozygous mice, orexin-A was 75% and orexin-B was 83% of wild type mice. While we did not measure orexin content in orexin-Flp mice, the orexin content in heterozygous mice should be decreased. Thus, the decreased duration in both heterozygous and homozygous mice compared with WT mice demonstrates that orexin is indispensable for sustaining the period of wakefulness. We added this limitation in our results to the Discussion as follows:

“However, the knock-in method has the disadvantage that orexin content in *OF* (KI/-) mice could be slightly decreased since Flp was knocked into the prepro-orexin locus. Chemelli et al. reported that orexin A content was decreased to 75% in heterozygous mice. (Chemelli et al., 1999). *OF* (KI/-) mice showed a shorter duration of wakefulness compared with WT control mice, and is likely due to the decrease in orexin.”

Given the reasons described above (that heterozygous mice were not normal due to a slight decrease in orexin), we compared the number of short, middle and long bouts of wakefulness between *OF* (KI/KI) homozygous mice (n = 7) and wild type mice (n = 8) during the dark period (Author response image 2). The wakefulness duration was divided into three categories, less than 5 min (short), 5 min to 1 hr (middle) and more than 1 hr (long) according to Stephenson et al., 2013. During the dark period (12 hr active period), *OF* (KI/KI) homozygous mice showed a significantly larger number of wakefulness episodes of short and middle durations compared with those of WT mice (short p = 0.0007 vs WT; middle p = 0.0001 vs WT; unpaired *t*-test). On the contrary, *OF* (KI/KI) homozygous mice exhibited no wakefulness episodes longer that 1 hr (p = 0.000001 vs WT, unpaired *t*-test). Taken together, these results strongly suggest that orexin is indispensable for sustained wakefulness.

**Author response image 2. respfig2:** Number of wakefulness bouts of short duration (~5 min), middle duration (5 min-1 hr) and long duration (1 hr~) in WT mice and *OF* (KI/KI) mice during the dark period. Data represent mean ± S.E.M.